# Variability in individual particle structure and mixing states between the glacier snowpack and atmosphere interface in the northeast Tibetan Plateau

Zhiwen Dong [a, b, *], Shichang Kang [a, c], Dahe Qin [a], Yaping Shao [b], Sven Ulbrich [b], Xiang Qin [a, d]

[a] State Key Laboratory of Cryosphere Sciences, Northwest Institute of Eco-Environment and Resources, Chinese Academy of Sciences, Lanzhou 730000, China;

[b] Institute for Geophysics and Meteorology, University of Cologne, Cologne D-50923, Germany;

[c] CAS Center for Excellence in Tibetan Plateau Earth Sciences, Beijing 100101, China;

[d] Qilian Shan Station of Glaciology and Ecological Environment, Chinese Academy of Sciences, Lanzhou 730000, China.

* **Corresponding Author. E-mail Address:** dongzhiwen@lzb.ac.cn (Z. Dong).

## Abstract

Aerosols affect the earth's temperature and climate by altering the radiative properties of the atmosphere. Changes in the composition, morphology structure and mixing states of aerosol components will cause significant changes in radiative forcing in the atmosphere. This work focused on the physicochemical properties of light-absorbing particles (LAPs) and their variability through deposition process from the atmosphere to the glacier/snowpack interface based on large-range observation in northeast Tibetan Plateau, and laboratory transmission electron microscope (TEM) and energy dispersive X-ray spectrometer (EDX) measurements. The results showed that LAPs particle structures changed markedly in the snowpack compared to those in the atmosphere due to black carbon (BC)/organic matter (OM) particle aging and salt-coating condition changes. Considerably more aged BC and OM particles were observed in the glacier and snowpack surfaces than in the atmosphere, as the concentration of aged BC and OM

varied in all locations by 4%-16% and 12%-25% in the atmosphere, respectively, whereas they varied by 25%-36% and 36%-48%, respectively, in the glacier/snowpack surface. Similarly, the salt-coated particle ratio of LAPs in the snowpack is lower than in the atmosphere. Albedo change contribution in the Miaoergou, Yuzhufeng and Qiyi Glaciers is evaluated using the SNICAR model for glacier surface distributed impurities. Due to the salt-coating state change, the snow albedo decreased by 16.7%-33.9% compared to that in the atmosphere. Such great change may cause more strongly enhanced radiative heating than previously thought, suggesting that the warming effect from particle structure and mixing change of glacier/snowpack LAPs may have markedly affected the climate on a global scale in terms of direct forcing in the Cryosphere.

**Keywords:** light-absorbing aerosols; atmosphere-snowpack interface; BC/OM particle structure aging; salt-coating change; particle internal mixing

## 1. Introduction

Aerosols affect the earth's temperature and climate by altering the radiative properties of the atmosphere (Jacobson, 2001, 2014; Ward et al., 2018). Snow cover and glaciers in cryospheric regions play an important role in global climate change because of their large areas of distribution on the earth's surface, especially in the Northern Hemisphere, e.g., in the Alpine Mountains, the Tibetan Plateau, northern hemisphere snowpack and the Polar Regions. Individual pollutant aerosols, e.g., black carbon (BC, or soot), organic carbon (OC) or organic matter (OM), mineral dust, deposited on glacier/snowpack surfaces cause enhanced surface heat absorption, acting as light absorbing particles (LAPs), and they thus impact radiative forcing in the cryosphere. Moreover, changes in composition, morphology structure and mixing states of different LAPs components will cause significant variability in individual particle radiative heating with largely varied surface albedo due to the changes in a single particle's mixing states (Cappa et al., 2012; Peng et al., 2016).

The Tibetan Plateau, acting as the "The Third Pole" region, is one of the largest cryosphere regions with a large ice mass beside the Polar Regions (Qiu, 2008). Large amounts of LAPs particles deposited on the glacier/snowpack surface can significantly

impact surface radiative forcing, and induce increased heat absorption of the atmosphere interface in lower and middle troposphere (Anesio et al., 2009; Kaspari et al., 2011; Dong et al., 2016, 2017), thereby causing rapid glacier melting in the region (Xu et al., 2009; Zhang et al., 2017; Skiles et al., 2018; Dumont et al., 2014).

Aerosols and climate interaction has become a major concern in the Tibetan Plateau region (Dong et al., 2016, 2017). For example, the long-range transport and deposition of BC (soot), various types of salts (e.g., ammonium, nitrate and sulfate), and aerosols, and their climate significance on the Tibetan Plateau glaciers have recently become heavily researched topics (Ramanathan et al., 2007; Flanner et al., 2007; Zhang et al., 2018). However, to date, notably limited studies have focused on the composition, mixing states, and change process of LAPs particles in the atmosphere-snowpack interface of the Tibetan Plateau glacier basins. Moreover, current modeling on cryospheric snow/ice radiative forcing's impact on climate change has rarely considered such influences from changes of the single particle's structure and mixing states (Ramanathan et al., 2007; Hu et al., 2018). Because of glacier ablation and LAPs accumulation in summer, the concentration of distributed impurities in glacier/snowpack surface is often even higher than that of the atmosphere (Zhang et al., 2017; Yan et al., 2016).

Therefore, this study aimed to provide a first and unique record of the individual LAPs' physicochemical properties, components variability and mixing states between the glacier/snowpack and atmosphere interface, based on aerosol (total suspended particle (TSP) on the aerosol filter) and the glaciers/snowpack surface-distributed impurity sampling in the northeast Tibetan Plateau during June 2016 to September 2017, to determine the LAPs particle's structure aging and mixing state changes through atmospheric deposition process from the atmosphere to the glacier/snowpack surface, thereby helping to characterize the LAPs' radiative forcing and climate effects in the Cryosphere region of Tibetan Plateau. Moreover, the albedo change contributions of LAPs in several glacier surfaces (e.g. Miaoergou, Yuzhufeng and Qiyi Glaciers) were evaluated using a SNICAR (Single-layer implementation of the Snow, Ice, and Aerosol Radiation) model for the salt mixing states of surface-distributed impurities of the observed glaciers. We organized the paper as follows: In section 2, we provided detailed

descriptions about data and method of individual aerosol particle sampling and analysis;
and in section 3 we presented the observed results and discussion of: (i) comparison of
LAPs components between atmosphere and snowpack interface; (ii) BC/OM particle
structure aging variability between atmosphere and snowpack interface; (iii) changes in
salt-coating conditions and BC/OM mixing states between the atmosphere and snowpack
interface; (iv) particle mixing states variability and its contribution to light absorbing. In
section 4, we concluded our results and also provided the future study objective for the
community.

## 2. Data and Methods

**Field Work Observation and Sampling.** The main methods of the study include the
fieldwork observations, and laboratory transmission electron microscope (TEM) and
energy dispersive X-ray spectrometer (EDX) instrument analysis. Atmospheric LAPs
samples (TEM aerosol filter samples) and the glacier/snowpack surface distributed
impurity samples were both collected across the northeastern Tibetan Plateau region in
summer between June 2016 and September 2017. Figure 1 shows the sampling locations
and their spatial distribution in the region, including locations in the eastern Tianshan
Mountains, the Qilian Mountains, the Kunlun Mountains and the Hengduan Mountains,
where large-range spatial scale observations were conducted (see Table 1). During the
fieldwork sampling, we used the middle-volume-sampler (DKL-2 (Dankeli) with a flow
rate of 150 L/min) for TEM filter sampling in this study, with a flow rate of 1 L min$^{-1}$
were used for TSP filter sampling in our study, by a single-stage cascade impactor with a
0.5 mm diameter jet nozzle and an airflow rate of 1.0 L min$^{-1}$. Each sample was collected
with 1-hour duration. After collection, the sample was placed in a sealed dry plastic tube
and stored in a desiccator at 25°C and 20±3% RH to minimize exposure to ambient air
before analysis, and particle smaller than 0.5 mm can be collected efficiently by the
instruments. In total, 80 aerosol samples were collected directly on the calcium-coated
carbon (Ca-C) grid filter. Additionally, 88 glacier/snowpack surface-snow samples were
collected on the glacier/snowpack surface (with 5 cm snow depth, each sampled for 200
mL) for comparison with the deposition process, and the snow samples are taken at the
same time of the atmospheric aerosol sampling. The aerosol/snow sampling method is
also the same to the previous study in Dong et al. (2016, 2017). The detailed information
on sampling locations, time period and aerosol/snow sample number are shown in Table
1. Snow samples were collected at different elevations along the glacier/snowpack
surfaces of the study. Pre-cleaned low-density polyethylene (LDPE) bottles (Thermo
scientific), stainless steel shovel, and super-clean clothes were used for the
glacier/snowpack surface-snow sample collection. All samples were kept frozen until
they were transported to the lab for analysis.
**TEM-EDX Microscopy Measurements.** Laboratory TEM-EDX measurements were
performed directly on the Ca-C filters grids (Dong et al., 2016). Ca-C grids were used as
filters with the advantage of clear and unprecedented observation for single-particle
analyses of aerosols and snowpack samples (Creamean et al., 2013; Li et al, 2014;
Semeniuk et al., 2014). Analyses of individual particle observations were conducted
using a JEM-2100F (Japan Electron Microscope) transmission electron microscope
operated at 200 kV. The analyses involved conventional and high-resolution imaging
using bright field mode, electron diffraction (Semeniuk, et al., 2014; Li et al., 2014), and
energy-dispersive X-ray spectrometry. A qualitative survey of grids was undertaken to
assess the size and compositional range of particles and to select areas for more detailed
quantitative work that was representative of the entire sample. This selection ensured that
despite the small percentage of particles analyzed quantitatively, our results were
consistent with the qualitative survey of the larger particle population on each grid.
Quantitative information on size, shape, composition, speciation, mixing state, and
physical state was collected for a limited set of stable particles. Some LAPs particles,
including nitrate, nitrite, and ammonium sulfate, though not stable under the electron
beam, can be well detected on EDX at low beam intensity. EDX spectra were collected
for 15 s in order to minimize radiation exposure and potential beam damage. All stable
particles with sizes 20 nm to 35 μm were analyzed within representative grid mesh
squares located near the center of the grid. Grid squares with moderate particle loadings
were selected for study to preclude the possibility of overlap or aggregation of particles
on the grid after sampling. The use of Ca-C grids resulted in clear and unprecedented
physical and chemical information for the individual particle types. Using TEM-EDX
microscope measurements, we can also easily derive the salt-coating conditions based on
the advantage of the transmission observation to obtain individual particle
inside-structure (Li et al., 2014). Particle (e.g. BC, OM) with salt coating will appear
clearly surrounded by various salts shell and with the BC/OM particle as the core. In
general, more than 400 particles were analyzed per grid; thus, more than 1200 particles
were analyzed from the three grid fractions per sample. Moreover, as the snow samples
melting will affect the individual particle composition during the measurements,
especially for various types of salts because the salt is unstable in high temperature (e.g.
Ammonium and Nitrates) and will change, thus the snow/aerosol samples were directly
observed under the TEM instrument and measured before it melted. All samples were
measured in frozen states.
**Snow Albedo Change Evaluation.** We also simulated the albedo change contributed by
individual particle mixing states' variability of LAPs. The SNICAR model can be used to
simulate the albedo of snowpack by the combination of the impurity of the contents (e.g.,
BC, dust and volcanic ash), snow effective grain size, and incident solar flux parameters
(Flanner et al., 2007). In this work, we use the online SNICAR model
(http://snow.engin.umich.edu/). In the SNICAR model, the effective grain sizes of snow
were derived from the stratigraphy and ranged from 100 μm for fresh clean snow to 1500
μm for aged snow and granular ice. The model was run with low, central, and high grain
size for each snow type to account for the uncertainties in the observed snow grain sizes.
Snow density varied with crystal size, shape, and the degree of rimming. The snow
density data used in the SNICAR model are summarized with low-, central-, and
high-density scenarios for the model run based on a series observations in the Tibetan
Plateau and previous literature (Judson and Doesken, 2000; Sjögren et al., 2007; Zhang et
al., 2018). In the model simulation, mineral dust (93.2±27.05 μg/g), BC (1517±626 μg/kg)
and OC (974±197 μg/kg) average concentration data, as well as other parameters, such as
effective grain size, snow density, solar zenith angle, and snow depth on the glaciers,
were all considered; The mass absorption cross-sections (MAC) for salt-coated BC was
referred to the average situation derived from the northern Tibetan Plateau glaciers
(Zhang et al., 2017, 2018; Yan et al., 2016; Wang et al., 2013). Though showing high
level, the BC concentration data used in this study is comparable to the previous work
results derived from the Himalaya ice core (Ming et al., 2008), as with relatively higher
average elevation in the Everest (its deposition site elevation 6500 m a.s.l. compared to
2900-4750 m a.s.l. of northeast Tibetan Plateau glacier sampling sites) and lower
atmospheric BC concentration. Besides, in this work we mainly focus on LAPs (BC, OC,
mineral dust, and others) in the glaciers and snowpacks for the surface distributed
impurities, thus impurity is often accumulated in summer with surface ablation and with
higher BC concentration.
When running the SNICAR model, BC/OM was assumed to be coated or non-coated with
sulfate (Flanner et al., 2007; Qu et al., 2014), or other salts. The mass absorption cross
section (MAC) is an input parameter for the SNICAR model; it is commonly assumed to
be 7.5 $m^2$/g at 550 nm for uncoated BC particles (Bond et al., 2013). For salt-coated BC
particles, the MAC scaling factor was set to be 1 $m^2$/g, following Qu et al. (2014) and
Wang et al. (2015). Other impurities (such as volcanic ash) were set to zero. In terms of
the albedo calculation, the BC and dust radiative forcing (RF) can be obtained by using
equation (1) (Kaspari et al., 2014; Yang et al., 2015):

$$\mathrm{RF} = \sum_{0.325\,\mu m}^{2.505\,\mu m} E(\lambda,\theta)(\alpha_{(r,\lambda)} - \alpha_{(r,\lambda,\mathrm{imp})})\Delta\lambda$$

193 (1)

196 where $\alpha$ is the modeled snow albedo with or without the impurities (imp) of BC and/or
197 dust; $E$ is the spectral irradiance (W $m^{-2}$); r is the snow optical grain size (μm); $\lambda$ is
198 wavelength (μm); and $\theta$ is the solar zenith angle for irradiance (°).

197 **3. Results and Discussion**

198 **3.1 Comparison of LAPs Components between Atmosphere and Snowpack Interface**

Figure 2 shows the component types of the individual LAPs particle found in the
atmosphere and glacier/snowpack of northeast Tibetan Plateau. Based on the above
microscope observations, aerosols were classified into seven components: NaCl salt,
mineral dust, BC (soot)/ fly ash, sulfates, ammonium, nitrates, and organic matter (OM).
Classification criteria of sampled particle types, mixing states and their possible sources
in the snow/atmosphere samples were indicated in Table S1. Figure 3 shows the
comparison of individual LAPs particle components types between glacier/snowpack and
atmosphere interface in northeast Tibetan Plateau region, which indicates the LAPs
composition in atmosphere of various locations as BC (mean percentage of 18.3%,
standard deviation (SD) 2.58), OC (28.2%, SD 3.49), NaCl (11%, SD 2.58), Sulfate (17%,
SD 3.49); Ammonium (4.8%, SD 3.01), Nitrate (7%, SD 2.83), Mineral dust (13.7%, SD
3.02), whereas the LAPs composition in glacier/snowpack surface as: BC (mean 21.3%,
SD 2.49), OC (31.2%, SD 2.44), NaCl (16.2%, SD 3.12), Sulfate (6.8%, SD 1.32),
Ammonium (2%, SD 0.81), Nitrate (3.3%, SD 0.95), Mineral dust (19.2%, SD 2.9). We
found that the impurity components show large differences between the snowpack and
atmosphere in all locations, implying significant change through the aerosols' deposition
processing in the interface (Figure 3). LAPs components have a large change of
proportion in the interface, probably due to different atmospheric cleaning rates and
atmospheric processing with dry/wet aerosol deposition. Sulfates and other salts in the
atmosphere act as salt-coating forms to other particles with aggregated states and will be
dissolved and taken away with precipitating snow and meltwater in the snowpack, which
will cause reduced salt components (e.g., sulfate, nitrate, NaCl, and ammonium) in the
glacier/snowpack surface compared to those in the atmosphere. Therefore, we can
observe obvious changes in composition and mixing states of the impurities between the
atmosphere and glacier/snowpack surface in Figure 3, as the ratio of BC, organic matter,
and mineral dust components in the snowpack increased greatly during this process,
whereas the ratio of various salts in the snowpack decreased significantly (Figure 3). The
change in morphology and structure will undoubtedly cause a significant variability of
impurities' heat absorbing property in both the atmosphere and the glacier/snowpack
surface, and such impacts will be discussed in a later section. Moreover, the deposition
flux and processing of various types of aerosol particles are different, causing the changes
in composition and mixing states of LAPs impurities between the atmosphere and
Cryosphere. Aerosol LAPs change during the atmospheric transport and deposition
processes (especially through wet deposition with precipitating-snow) will mainly lead to
large variability of individual particle's structure and morphology; for example, the
particle's aging, salt-coating, and mixing states changes of BC and organic matter (with
internal or external mixing), as indicated in following sections, which will cause further
influences on radiative forcing of the glacier/snowpack surface as discussed in section

236   3.4.

**3.2 BC/OM Particle Aging between Atmosphere and Snowpack Interface**

Figure 4 shows how the particle's structure changes during the individual particle aging
process when deposited from the atmosphere onto the glacier snowpack surface. Figure
4a-4d is the representative particles of fresh BC/OM with fractal morphology and a large
amount in the atmosphere, whereas Figure 4e-4h is the representative particles of aged
BC/OM with aggregated spherical morphology in the glacier/snowpack surface. It is clear
that abundant aerosol particles were observed with relatively fresh structure in the
atmosphere, similar to previous studies (e.g., Li et al., 2015; Peng et al., 2016). As shown
in Figure 4a-4d, the fresh aerosol particles of BC and OC (or organic matter, OM)
appeared very common in the atmosphere as the main parts, whereas as shown in Figure
4e-4h, more aged particles were found deposited in the glacier/snowpack surface. This
process is characterized by the initial transformation from a fractal structure to spherical
morphology and the subsequent growth of fully compact particles. Previous work has
indicated the structure and mass absorption cross (MAC) section change of BC particles
in the atmosphere (Peng et al., 2016; Yan et al., 2016), but did not discuss such change
phenomena of OM particles' change during the structure-aging process. This study
reveals clearly the structure and morphology change of BC and OM particles' structure
aging through the transport and deposition process to the glacier snowpack from the
atmosphere (Figure 4).
Based on TEM-EDX observations, we evaluated the aged BC/OM particle composition
ratio (%) in the snowpack and the atmosphere, respectively. Figure 5 shows the aging of
BC/OM individual particles and their composition ratio (%) change with the deposition
process from the atmosphere to the glacier/snowpack surface. Figure 5 indicates that in
atmosphere the composition ratio is as fresh BC (mean percentage of 29.7%, with SD
3.95), fresh OC (41.8%, 4.34), aged BC (9.8%, 4.02), and aged OC (18.7%, 4.11); while
in the snow the composition ratio is as fresh BC (mean percentage of 8.4%, SD 2.71),
fresh OC (17.7%, 4.42), aged BC (31.5%, 2.99), and aged OC (42.4%, 4.45). The

proportion of aged BC and OM particles varied from 4%-16% and 12%-25% in the atmosphere, respectively, and varied from 25%-36% and 36%-48% in the glacier/snowpack surface, respectively. The amount of aged particles in snowpack is 2-3 times higher than that in the atmosphere. In the atmosphere, the BC/OM both showed high ratios of fresh structure particles (fractal morphology), while in the glacier/snowpack surface more particles indicated aged structure (spherical morphology), although there was a small portion of particles still fresh (Figure 5). The change proportion of BC/OM particle aging is very marked between the snow and the atmosphere. The particle structure is a very important factor influencing light absorbing (Peng et al., 2016); thus, such changes in BC/OM particles' structure aging between the glacier snowpack and atmosphere will actually influence the total heat absorbing of the mountain glacier/snowpack, even affecting that of the whole cryosphere on earth's surface.

**3.3 Changes in Salt-Coating Conditions and BC/OM Mixing States**

In addition to particle structure aging, we find evident variability in particle salt-coating conditions between the atmosphere and glacier/snowpack interface during the observation period (Figure 6). Figure 6 demonstrates the different salt-coating examples for individual aerosol particles (including BC, OM, and mineral dust) in the atmosphere in various glacier basins in the northeast Tibetan Plateau. We found that the salt-coating form is very common for impurity particles in the atmosphere, which will, of course, cause a significant influence on radiative forcing of the atmosphere. A large part of fresh BC/OM (with fractal morphology) and mineral dust particles were coated by various salts, such as sulfate, nitrates, and ammonium. Such obvious salt-coating conditions will cause reduced atmospheric radiative forcing, due to the increase of albedo (IPCC, 2013).

Similarly, we also evaluated the salt-coated particle ratio for BC/OM and its change between glacier/snowpack and atmosphere (Figure 7). Figure 7 shows the salt-coating proportion of impurity particles and its difference between the glacier/snowpack and atmosphere interface at those locations. In Figure 7, the salt-coated particles in atmosphere accounted for mean ratio of 54.61% (with SD 12.02) in various locations, while that in the snow of the glacier/snowpack was 18.59% (with SD 7.04). The

proportion of salt-coating particles varied largely from the atmosphere to the
glacier/snowpack surface (2-4 times more in the atmosphere than that in snow). The
change proportion of salt-coating particles is very marked, and this change will cause
very complicated changes in a particle's mixing states and structure.
Figure 8 shows the situation of internal mixing states of BC (soot), organic matter (OM)
and mineral dust particles in various glacier snowpacks in the region, which demonstrates
the influence of the transport and deposition process to a particle's structure change. Most
salts in the salt-coated particles will disappear when deposited into the glacier/snowpack
surface, and the mixing states change largely to the internal and external mixing forms
with BC/OM as the core. The proportion change of an internally mixed BC particle with
other particles is presented in Figure 9, showing great increases in internal mixing after
deposition among the locations in the whole northeast Tibetan Plateau region. As shown
in Figure 9, the internally mixed particles of BC in atmosphere accounted for mean ratio
4.68% (with SD 3.07) in various locations, whereas that in the snow of glacier/snowpack
was 14.85% (with SD 4.93). We find that with the salt-dissolution, a large part of LAPs
particles changed to the internally mixed BC/OM particle with other aerosol particles. As
a large number of particles lose the salt coating in the snowpack compared with those in
the atmosphere, the whole process will certainly increase the heating absorption
proportion of the LAPs. Moreover, as shown in Figure 10, average conditions of single,
internally and externally mixed BC/OM individual particles in the glacier/snowpack of
the northeast Tibetan Plateau changed greatly with the diameter of the particle. In Figure
10, the mixings states of BC/OC in the glacier/snowpack snow of northeast Tibetan
Plateau showed that the internally, single and externally mixed BC/OC particles
accounted for mean ratio of 69.2% (SD 22.5), 5.35% (SD 1.72), and 25.95% (with SD
22.4), respectively. With the increase in particle size, most BC/OM particles (PM>1 μm)
showed internal mixing conditions, which will influence the RF of the glacier snowpack.
**3.4 Particle Mixing States Variability and Its Contribution to Light Absorbing**
Additionally, the extent of influence of LAPs' particle mixing state changes are also
important and need to be evaluated for radiative forcing. The SNICAR model is often
employed to simulate the hemispheric albedo of glacier/snowpack for a unique
combination of LAPs contents (e.g., BC, dust, and volcanic ash), snow effective grain
size, and incident solar flux characteristics (Flanner et al., 2007). We also evaluated the
influence on albedo change caused by individual particle structure and mixing state
changes in the glaciers/snowpack of the northeast Tibetan Plateau region. Figure 11
showed the evaluation of snow albedo change of BC-salt coating change in the snowpack
compared with that in the atmosphere using SNICAR model simulation in the MG, YG,
QG, showing the albedo change of snow surface impurities in snowpack compared to that
of the atmosphere. The parameters input for SNICAR model have been described in the
method section. Mineral dust, BC and OC average concentration data, as well as other
parameters, such as effective grain size, snow density, solar zenith angle, and snow depth
on the glaciers, and MAC for BC were referred from the average situation in previous
work of northern Tibetan Plateau glaciers (Zhang et al., 2017, 2018; Yan et al., 2016;
Wang et al., 2013). As shown in Figure 12, the surface albedo in MG, YG, and QG
decreased by 16.7%-33.9% caused by salt-coating changes, when compared to that of the
hypothetical similar situation of impurities' composition as that in the atmosphere. Based
on the LAPs salt-coating-induced albedo changes, RF was calculated by equation (1) for
the different scenarios. The results show that the RF change caused by salt coating
changes, varied between 1.6–26.3 W m$^2$ depending on the different scenarios (low,
central, and high snow density), respectively.
Figure 12 shows a schematic diagram model for the explanation of the particle structure
aging and salt-coating changes, and its total influence to the radiative forcing between the
atmosphere and glacier/snowpack interface on the northeast Tibetan Plateau. From the
above discussion, we find a clear variability in LAPs particles' mixing forms between the
glacier/snowpack surface and atmosphere, mainly originating from the morphology
changes of the LAPs particle's structure (e.g. aging of BC/OM), and salt-coating changes
from increased internal mixing of BC/OC particles, as many particles without salt-coating
will change to internal mixing with BC/OM particles as a core, or external mixing with
BC/OM, which will also significantly influence the total RF of the mountain
glaciers/snowpack in the Cryosphere as indicated in previous work (Jacobson et al.,
2001). Moreover, due to glacier ablation and accumulation of various types of impurities,
the concentration of impurities in the snowpack surface is often even higher than that of
the atmosphere (Zhang et al., 2017; Yan et al., 2016).
In general, as shown in Figure 12, (ⅰ) more fresh BC/OM particles were observed in the
atmosphere, whereas more aged BC/OM particles were found on the glacier/snowpack
surface. Aged BC/OM particles often mean stronger radiative forcing in the snowpack
than in the atmosphere (Peng et al., 2015). (ⅱ) More salt-coated particles were found in
the atmosphere of the glacier basin, whereas reduced salt coating was found in the
glacier/snow surface. With thick salt coating, the LAPs' light- absorbing properties may
not be that much stronger than the particles without coating, as most salts (sulfate,
nitrates, ammonium, and NaCl) did not have strong forcing because of their weak light-
absorbing property and high hygroscopicity in the mixing states (IPCC, 2013; Li et al.,
2014), especially for sulfate/nitrate aggregated particles. (ⅲ) With the salt-coating
decrease, more internally mixed particles of BC/OM surrounded by a well-mixed
salt-shell were observed from the individual particles of LAPs in the snow-ice of the
cryospheric glacier basin when compared to that of the atmosphere. Internally mixed
particles of BC/OM have shown the strongest light absorption in previous modeling
studies (Cappa et al., 2012; Jacobson et al., 2000), as BC acts as a cell-core particle with
organic matter particles (also sometimes including some salts) surrounded. In previous
study the mixing state was found to affect the BC global direct forcing by a factor of 2.9
(0.27 $Wm^{-2}$ for an external mixture, +0.54 $Wm^{-2}$ for BC as a coated core, and +0.78 $Wm^{-2}$
for BC as well mixed internally) (Jacobson, 2000), and that the mixing state and direct
forcing of the black-carbon component approach those of an internal mixture, largely due
to coagulation and growth of aerosol particles (Jacobson et al., 2001), and also found
radiative absorption enhancements due to the mixing state of BC as indicated in Cappa et
al. (2012), and He et al.(2015). (ⅳ) In addition to the light-absorbing from the above
particle structure change, the absorbing property of some components in the atmosphere
and cryosphere (snow and ice) also show a large variability, as most mineral and OM (or
OC) particles show negative radiative forcing in the atmosphere while showing positive
forcing in the glacier/snowpack surface, as indicated from IPCC AR5 (2013), Yan et al.
(2016), Zhang et al. (2018), and Hu et al. (2018). Thus, the light-absorbing of LAPs as a
whole will increase greatly in glacier/snowpack surface environments.

Therefore, the great change of glacier/snowpack surface albedo may cause more strongly enhanced radiative heating than previously thought, suggesting that the warming effect from particle structure and mixing change of glacier/snowpack LAPs may have markedly affected the climate on a global scale in terms of direct forcing in the Cryosphere.

## 4. Conclusions

The results showed that the LAPs particle structure changed greatly in snowpack compared to that in the atmosphere, mainly due to particle aging (mainly BC and organic matter), and the salt coating reduction process through the impurity particle's atmospheric deposition. Much more aging BC and OM and more internally mixed BC particles were observed in glacier snowpack than in the atmosphere during the simultaneous observations; for example, the proportion of aged BC and OM varies from 4-16 % and 12-25% in the atmosphere respectively, and varies from 25-36% and 36-48% respectively in the snowpack of the cryosphere. In addition to the heat absorbing from the above LAPs particle structure change, the absorbing property of dust and OC in atmosphere and cryosphere (snow and ice) also shows a large difference.

A schematic diagram model shown in the figure linking the explanation the LAPs' structure aging and salt-coating change and comparing their influences to the radiative forcing between the atmosphere and glacier snowpack was presented in the study. The LAPs in glacier/snowpack will change to more aged and internally mixed states compared to that of the atmosphere. Thus, the light absorption of the LAPs as a whole will increase greatly in glacier snowpack environments. Moreover, we also evaluated the increase in radiative forcing caused by LAPs particle structures and mixing state changes. The albedo changes in MG, YG and QG were evaluated using the SNICAR model simulation for distributed surface impurities in the observed glaciers caused by salt coating changes, which decreased by 16.7%-33.9% compared to glacier surface with similar conditions as in the atmosphere. The RF change caused by salt coating changes, varied between 1.6–26.3 W m$^2$ depending on the different scenarios (low, central, and high snow density), respectively. We find that the LAPs-individual-particle related albedo and radiative forcing change in this work is of importance in understanding the contribution of individual particle structure and mixing change in atmosphere-snowpack

interface, which may have markedly affected the climate on a global scale in terms of direct forcing in the Cryosphere, and need to be further studied in future.

## Acknowledgements

This work was funded by the National Natural Science Foundation of China (41671062, 41721091), the State Key Laboratory of Cryosphere Sciences (SKLCS-ZZ-2018), and the Youth Innovation Promotion Association, CAS (2015347). We also thank the field work team (especially to Li G., Li Y. and Chen S F.) in the northeast Tibetan Plateau for their logistical work and sample collections. All the data used are contained within the paper and tables, figures, and references.

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

**Tables**
**Table 1. Sampling locations, sampling dates, and cryoconite-snow depth at mountain**
**glaciers of the northeast Tibetan Plateau**

| Sites | Glacier | Mountains | Locations | Altitude (m a.s.l.) | Sampling Date | Number Snow/Aerosols | Particles Calculated |
|-------|---------|-----------|-----------|---------------------|---------------|----------------------|----------------------|
| MG | Miaoergou Glacier | Tianshan Mountains | 42.59°N, 94.16°E | 3800-4200 | 12-13 June 2017 | 8/8 | >1200 |
| LG12 | Laohugou Glacier No.12 | Qilian Mountains | 39.20°N, 96.34°E | 4300-4700 | 10-25 July, 2016, 3-8 June, 10-21 August 2017 | 20/24 | >1200 |
| QG | Qiyi Glacier | Qilian Mountains | 39.14°N, 97.45°E | 4200-4750 | 10-12 June 2017 20-22 August 2017 | 11/8 | >1200 |
| DS | Daban Snowpack | Daban Mountains | 37.21°N, 101.24°E | 3500-3700 | 3-4 June 2017 | 8/4 | >1200 |
| LG | Lenglongling Glacier | Qilian Mountains | 37.51°N, 101.54°E | 3558-3990 | 5-7 June 2017 | 12/5 | >1200 |
| SG | Shiyi Glacier | Qilian Mountains | 38.21°N, 99.88° E | 3900-4400 | 3-4 June 2017 | 9/6 | >1200 |
| YG | Yuzhufeng Glacier | Kunlun Mountains | 35.41°N, 94.16°E | 4300-4720 | 12 June 2017 | 12/11 | >1200 |
| GS | Gannan Snowpack | Gannan Plateau | 34.2°N, 103.5°E | 2900-3200 | 4-8 May 2017 6-9 August 2017 | 6/6 | >1200 |
| DG | Dagu Glacier | Hengduan Mountains | 33°N, 101°E | 3200-3900 | 20-22 Sept 2017 | 2/3 | >1200 |
| HG | Hailuogou Glacier | Hengduan Mountains | 31°N,101°E | 2900-3500 | 11-12 August 2017 | 6/4 | >1200 |







## Figure Captions

**Figure 1** Location map showing the sampled glaciers and snowpack in the northeast Tibetan Plateau, including the Miaoergou Glacier (MG), Laohugou Glacier No.12 (LG12), Qiyi Glacier (QG), Lenglongling Glacier (LG), Shiyi Glacier (SG), Dabanshan snowpack (DS), Yuzhufeng Glacier (YG), Gannan Snowpack (GS), Dagu Glacier (DG), and Hailuogou Glacier (HG), where large-range field observations of atmosphere and glacier surface impurities were conducted.

**Figure 2** Component types of individual haze particles in northwest China. Based on the above microscope observation, aerosols were classified into seven type components: NaCl salt, mineral dust, fly ash, BC (soot), sulfates, nitrates, and organic matter.

**Figure 3** Comparison of individual particles' compositions of light-absorbing impurities in the (a) atmosphere and (b) glacier/snowpack surface in the northeast Tibetan Plateau, and (c) a photo of snowpack and glaciers in the Qilian Mountains taken from flight in autumn 2017, showing large distribution of snow cover and glaciers in the north Tibetan Plateau region -round.

**Figure 4** Structure change during the aging of individual black carbon (BC) / organic matter (OM) particles when deposited from the atmosphere onto snow and ice surface. Figures 4a-4d is representative of atmosphere, while Figure 4e-4h shows the condition of glacier/snowpack.

**Figure 5** LAPs aging of BC/OC individual impurity particles and composition ratio (%) change during the deposition process from the atmosphere to glacier snowpack, in the figure (a) is the atmosphere, and (b) is the snowpack.

**Figure 6** Examples of different salt-coating conditions of BC, OM and dust for individual particles in the atmosphere of various glacier basins in northeast Tibetan Plateau

**Figure 7** Salt-coating proportion changes of individual impurity particles between glacier snowpack and atmosphere in various locations of northeast Tibetan Plateau

**Figure 8** Internal mixing states of BC (soot) and OM in the various glacier snowpack in

northeast Tibetan Plateau in summer 2016-2017
**Figure 9** The proportion change of internally mixed BC particle with other particles,
showing the obvious increase of internally mixed BC/OM in glacier snowpack compared
with those in the atmosphere in summer 2016-2017
**Figure 10** Average conditions of single, internally and externally mixed BC/OM
individual particles in the snowpack of northeast Tibetan Plateau glaciers, showing most
of the BC/OM with diameter >1 μm in internally mixing conditions.
**Figure 11** Evaluation of snow albedo change of BC-salt coating change in the snowpack
compared with atmosphere using SNICAR model simulation in the MG (a, b), YG (c, d),
LG 12 (e, f), which shows the largely decreased albedo of snow surface impurities in
snowpack compared to that of the atmosphere, implying markedly enhanced radiative
forcing in the snowpack surface impurities.
**Figure 12** Schematic diagram linking aging and salt coating change and comparing its
influence to the radiative forcing between the atmosphere and snowpack of a remote
glacier basin, thereby causing markedly enhanced heat absorption.












**Figure 1**

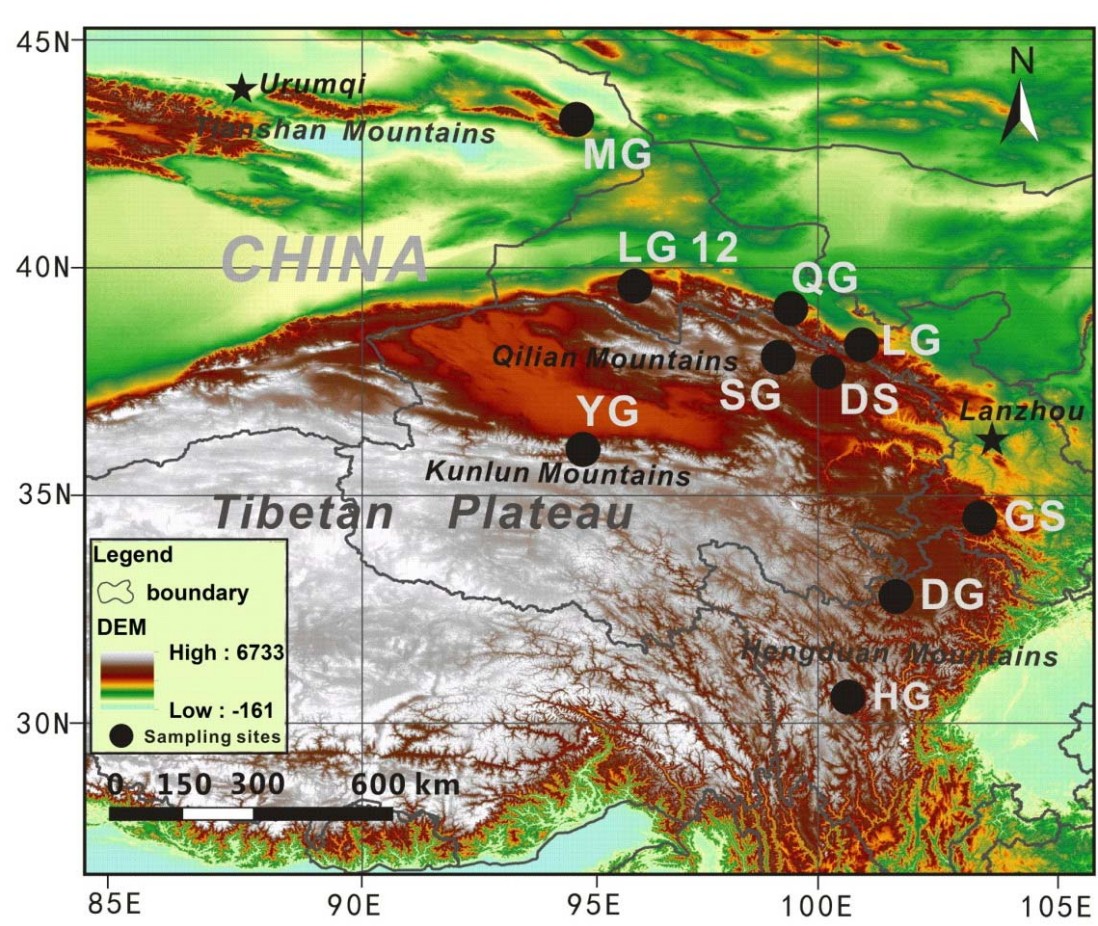



**Figure 2**

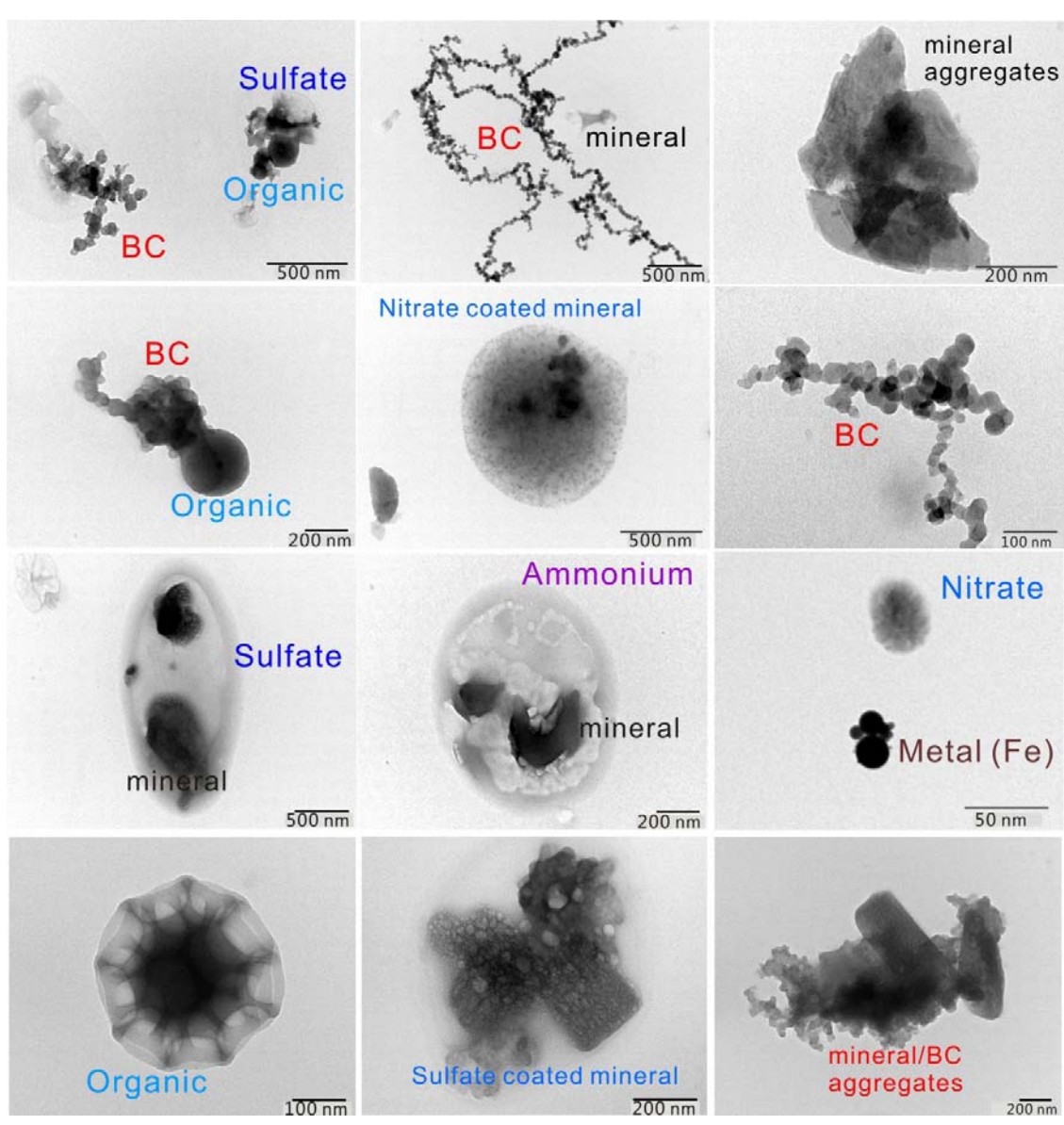





**Figure 3**

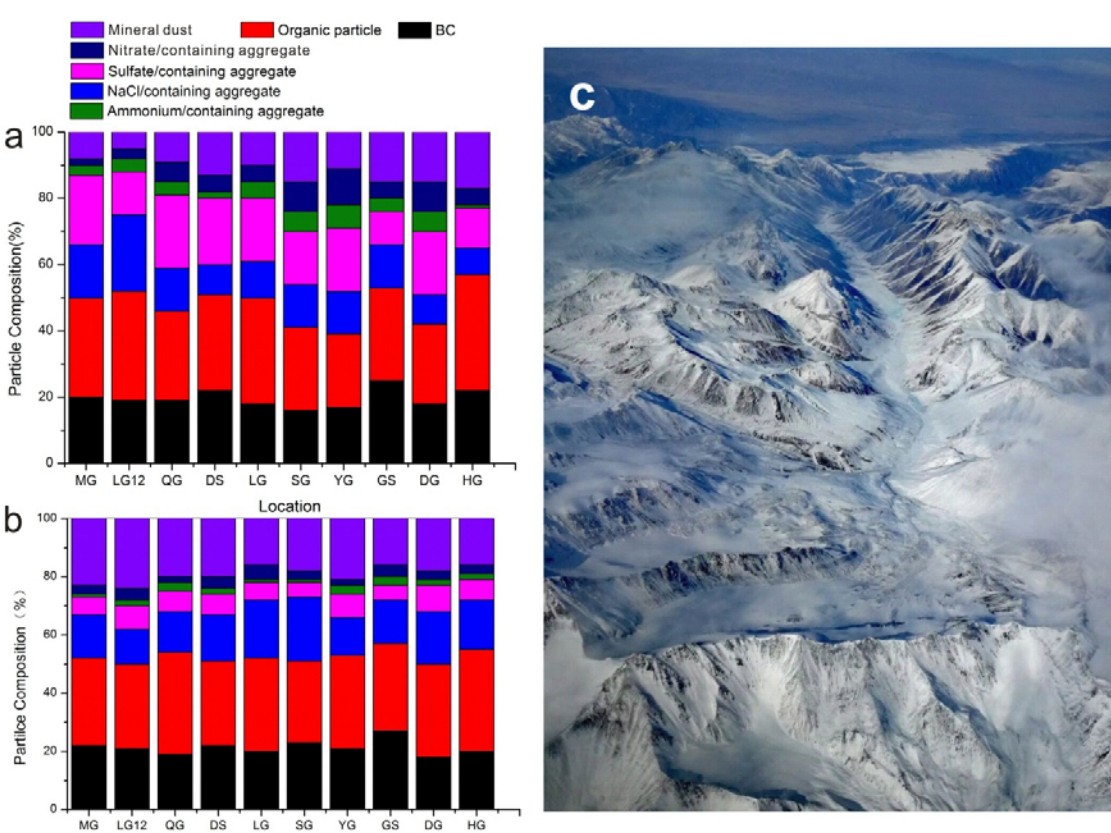







**Figure 4**

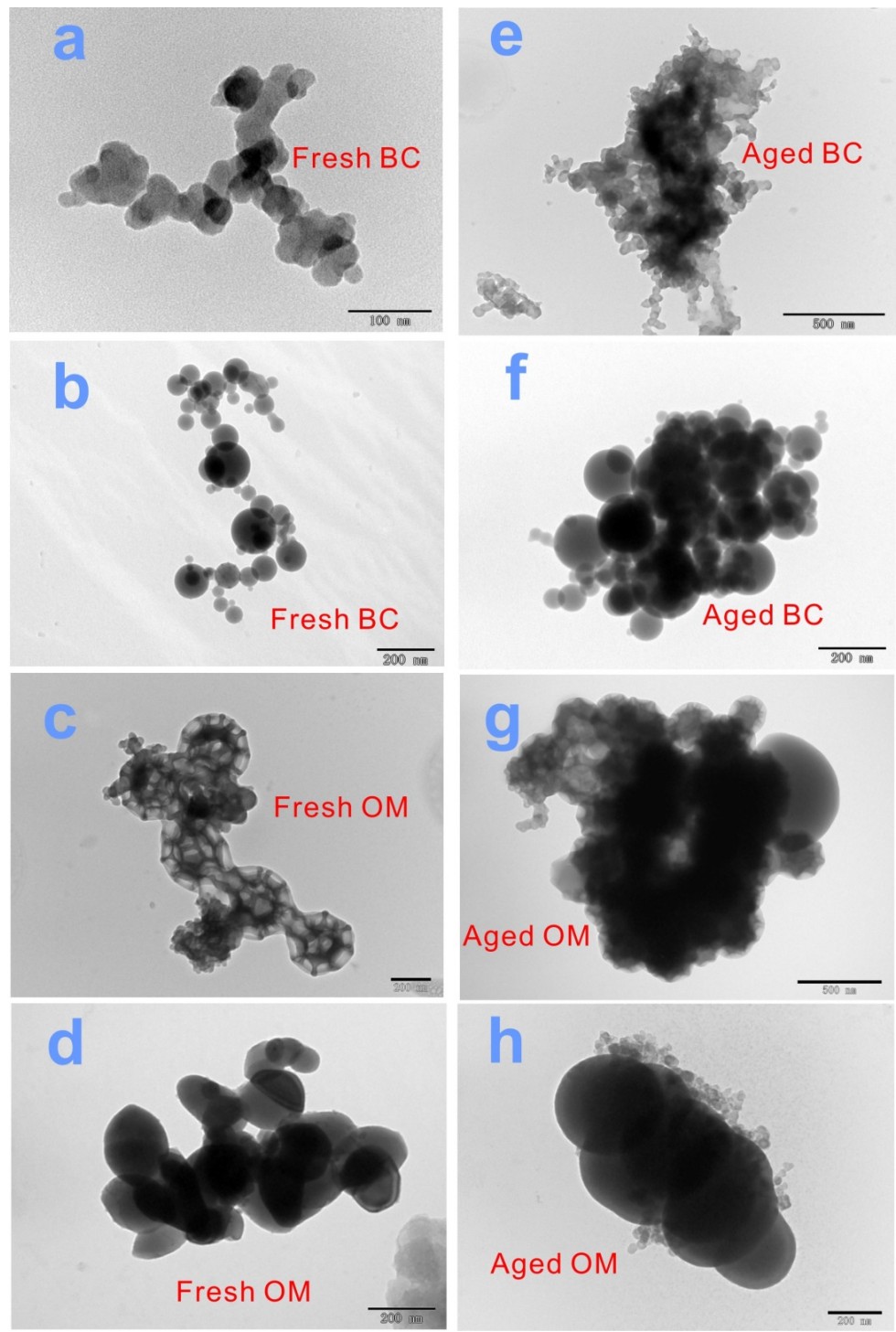


**Figure 5**

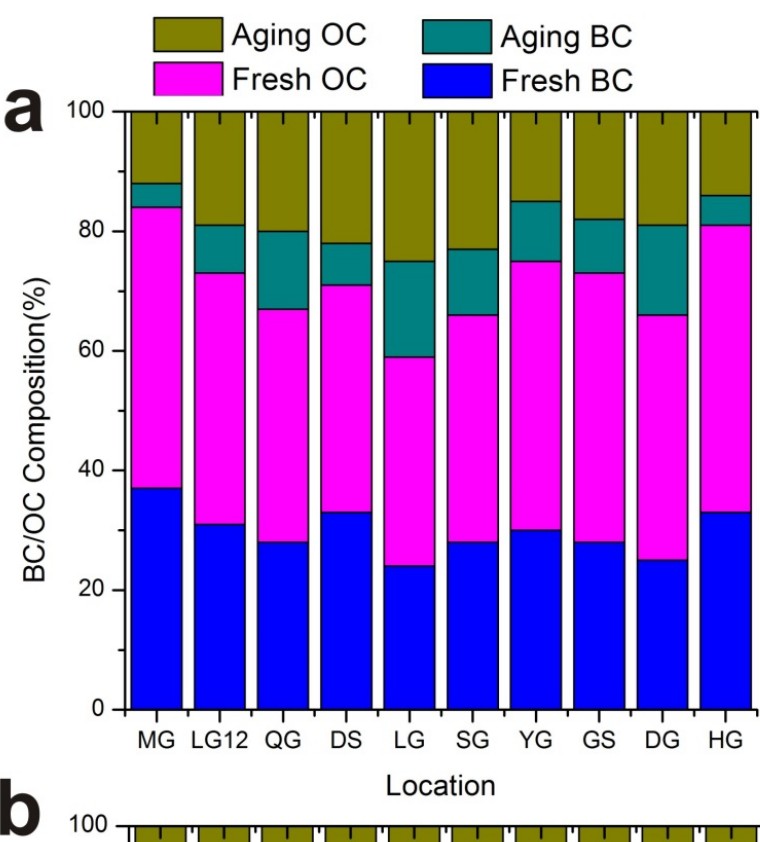

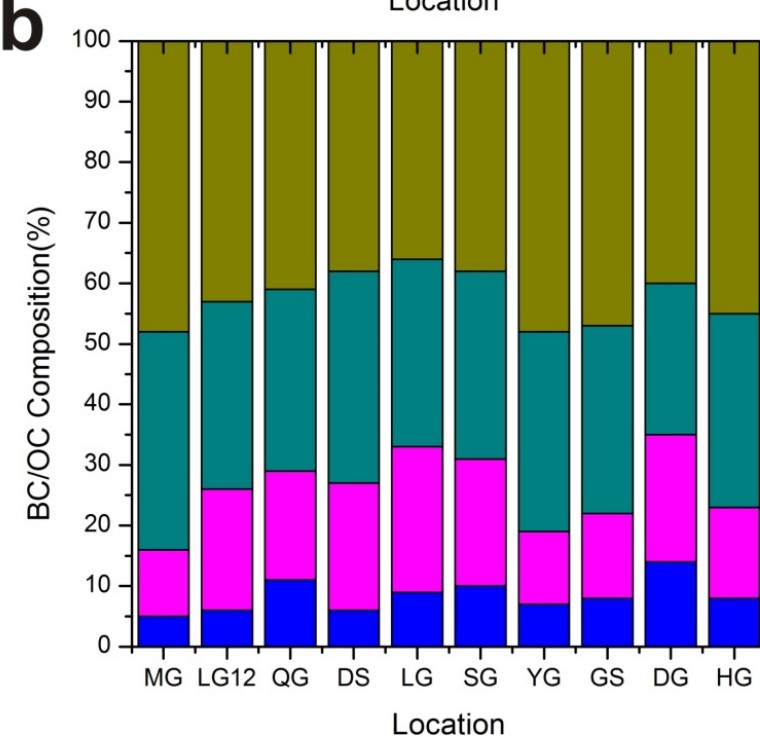


**Figure 6**

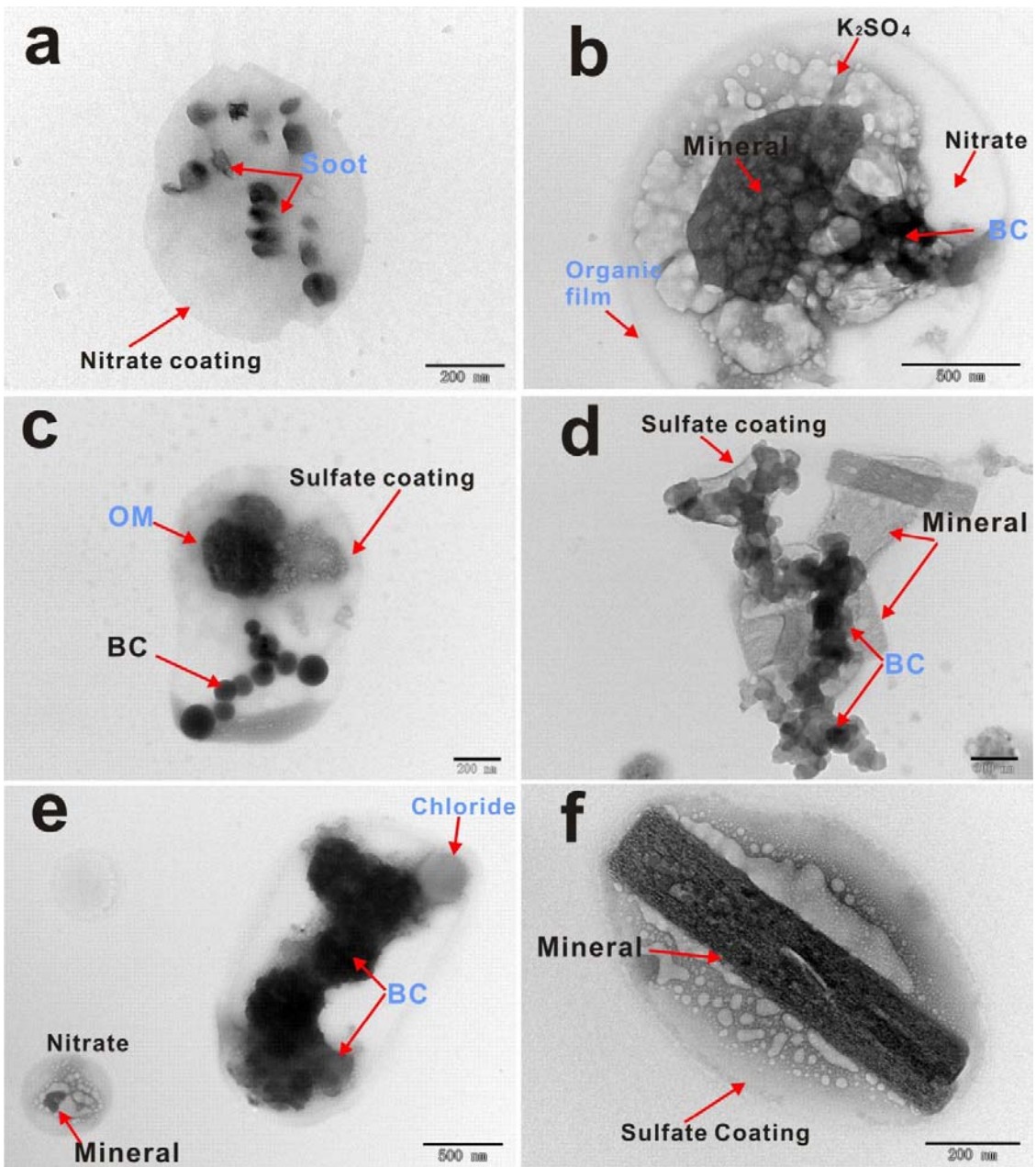





**Figure 7**

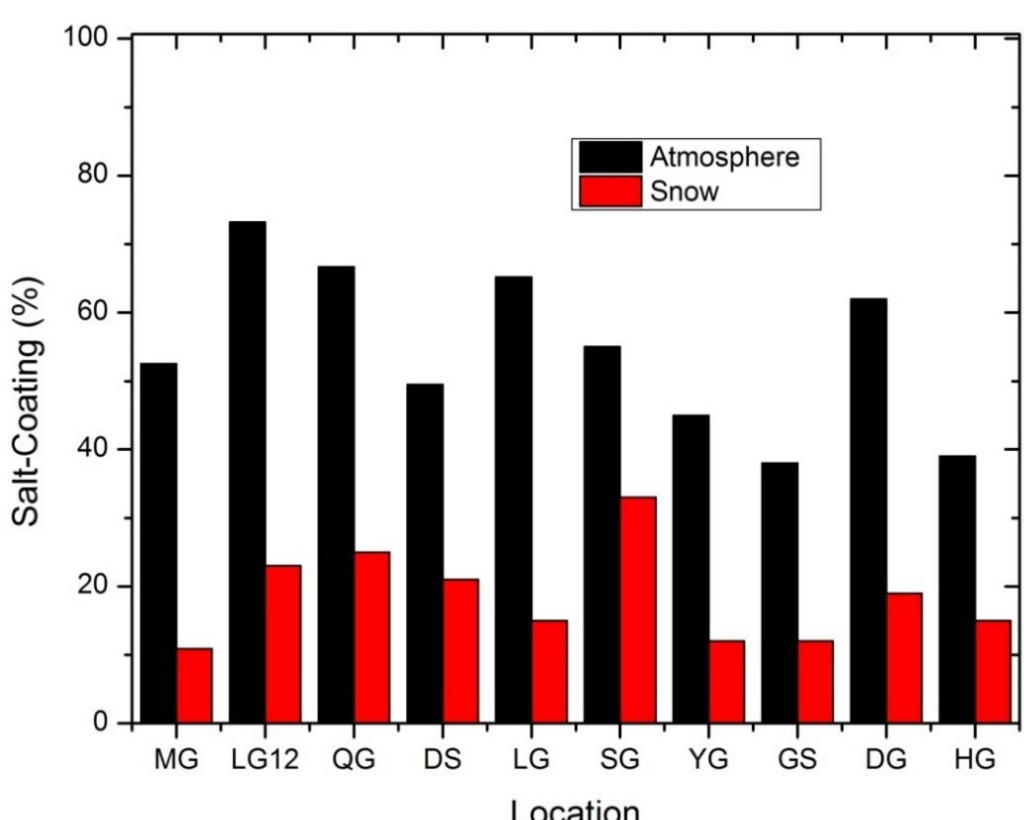




**Figure 8**

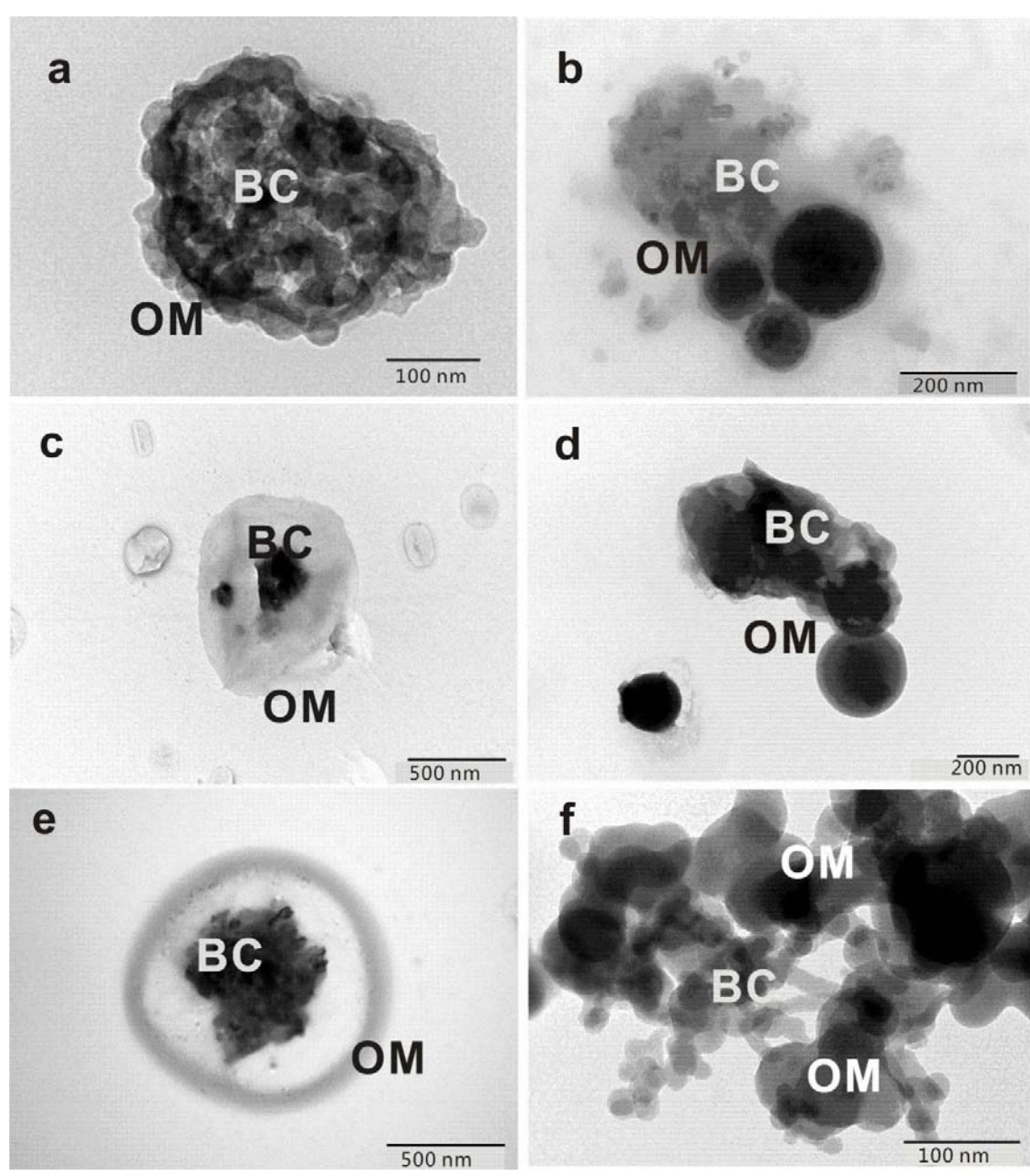





**Figure 9**

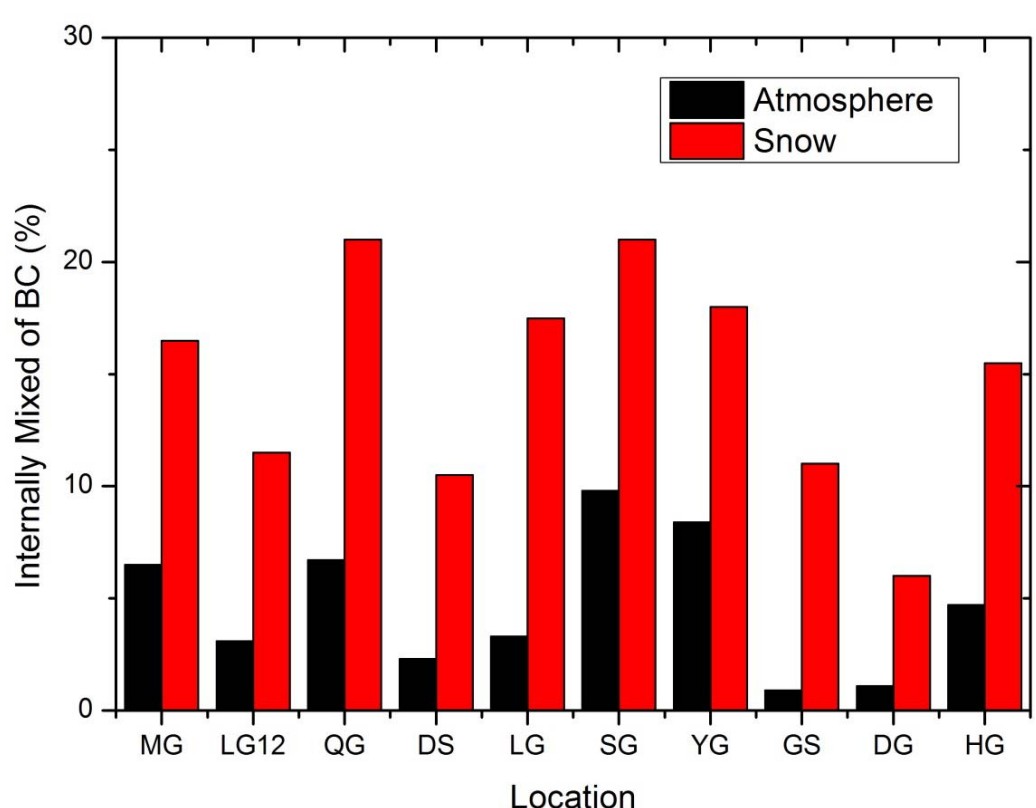










**Figure 10**

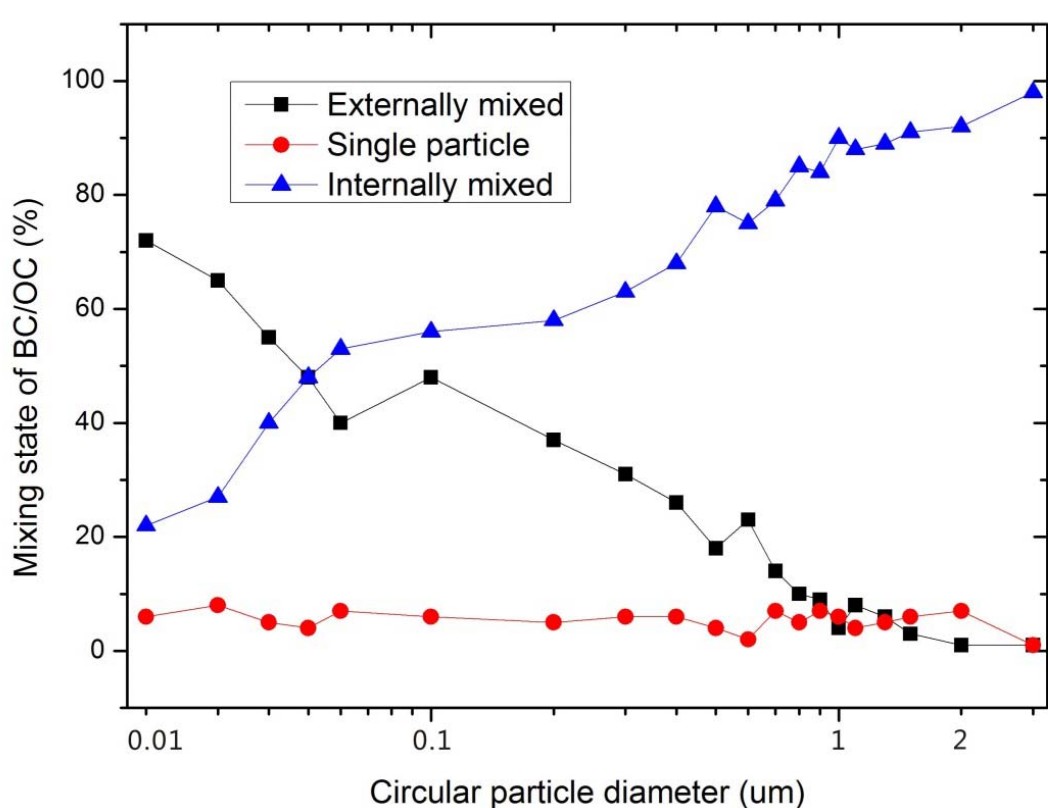














**Figure 11**

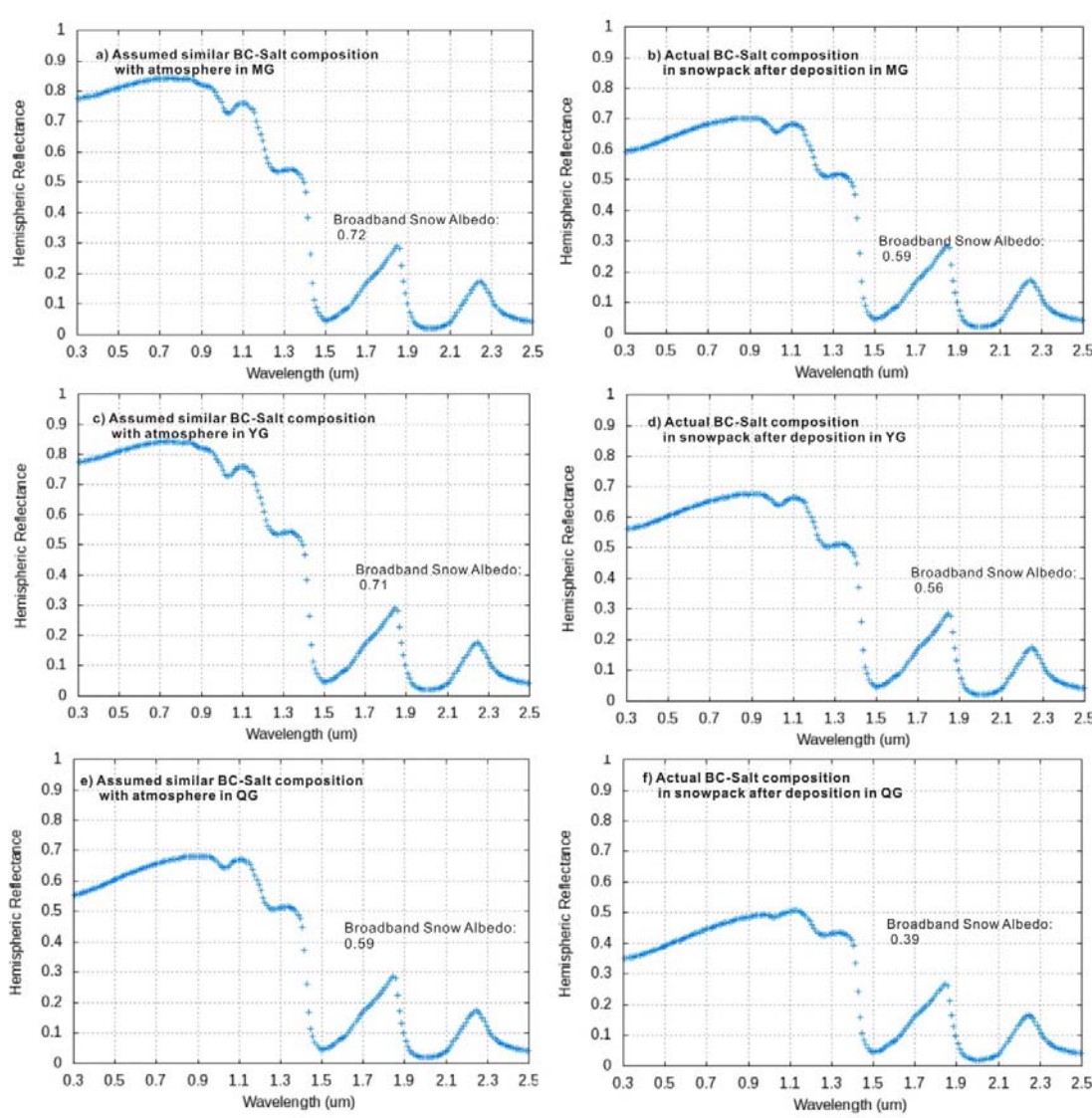








**Figure 12**


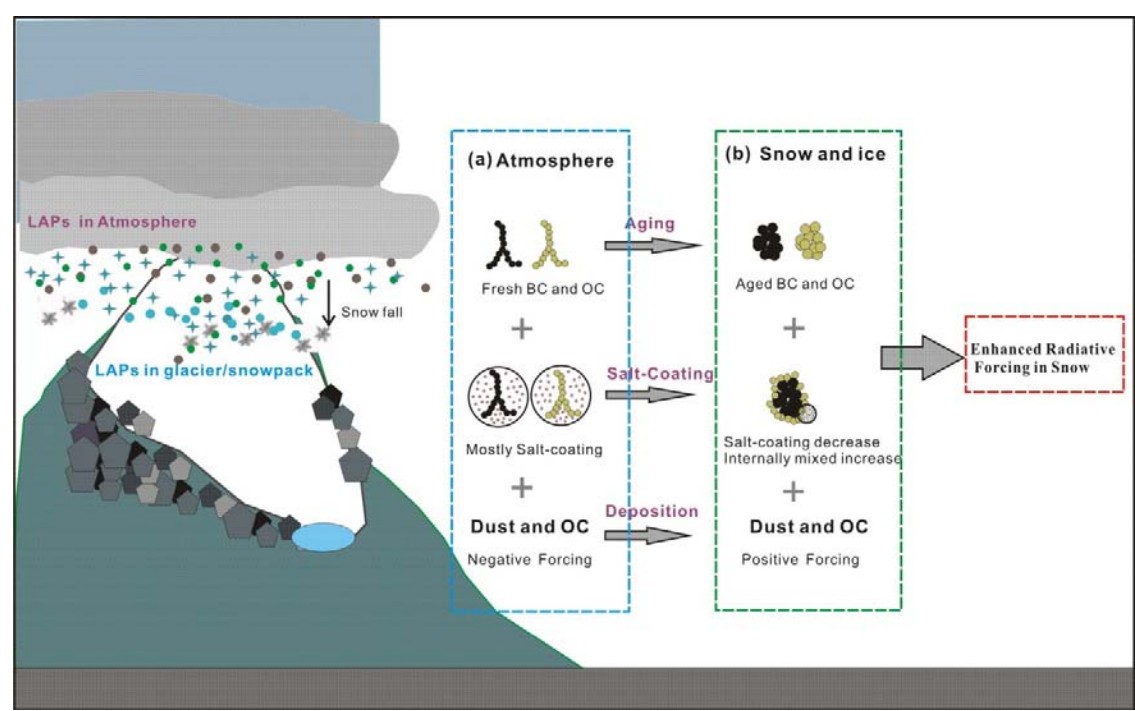

