# Peer review of "Variability in individual particle structure and mixing states between the glacier snowpack and atmosphere interface in the northeast Tibetan Plateau"

_The Cryosphere, 2018_

## Referee Comment (RC1) · Anonymous Referee #1 · 4 Oct 2018

General statement: This is an interesting study suitable for publication in TC. There are some important revisions required, but they should be easy for the authors to make.

Major comment:

The large radiative forcing shown in Figure 11 (and cited in the abstract) is caused by the huge amounts of impurities in these glaciers. The authors need to point out that these glaciers are much more polluted than is normal for Tibetan glaciers. Line 120 gives the average value of BC (for 10 glaciers) as 854 ppm, or 854,000 ppb, i.e. about a factor of 40,000 larger than the amounts reported for Tibetan glaciers by Ming et al 2008 (ACP 8, 1343-1352). Some discussion is required before we can believe the

results of this paper. I am also surprised that [BC] is ∼10x [MD]. Previous reports find more MD than BC in Tibet.

Minor Comments:

line 43. "various salts . . . cause enhanced surface heat absorption". Which salts do you mean? Most salts are non-absorptive at UV, visible, and near-infrared wavelengths.

line 51. Delete "et al"

line 54. Anesio et al 2009 is missing from the reference list. Kaspari et al 2011 is also missing.

line 55. Xu et al. is missing from the reference list.

line 61. McConnell et al. 2007 is missing from the reference list.

line 74. Define TSP.

line 105. Semeniuk et al. 2014 is missing from the reference list.

line 120. Please give the values of MD, BC, OC for each individual glacier. Put them in Table 1.

line 169. "Previous work". Give a reference.

Table 1. The altitude for DF is given as 390 m. Probably you mean 3900 m.

Table 1. Add three more columns, giving the concentrations of BC, OC, MD in the surface snow of each glacier.

line 398 (Figure 2 caption) "nitrates". The legends in Figures 2 and 6 say nitrite not nitrate.

line 400 (Figure 3 caption). "snow and ice". Which of the ten sites were snow; which sites were ice?

line 414 (Figure 8 caption) "mineral dust particles". No particles in Figure 8 are labeled

as mineral dust.

Figure 2. Labels on the scale bars are illegible. Increase the font size.

Figure 10. The vertical axes for the three graphs should all use the same scale, for easy comparison by the reader.

Figure 10 legend. change "Mixted" to "Mixed".

Figure 12. The listed values for broadband albedo have too many significant figures. For example change "0.29774863" to "0.30".
* * *

---

## Referee Comment (RC2) · M. Dumont (Referee) · 8 Oct 2018

**Review of "Variability in individual particle structure and mixing states between glacier snowpack and atmosphere interface in the northeast Tibetan Plateau " by Dong et al.**

**Summary**

This paper presents a dataset that explores the physical and chemical properties of light absorbing particles (LAPs) in the atmosphere and in the surface snowpack at several places in the Tibetan Plateau. Observations from TEM and EDW measurements are described. A tentative scheme to explain the observations is proposed along with an assessment of the changes in radiative impact.

**Recommendations**

This is a really interesting, rich and fascinating dataset, the conclusions drawn by the authors are of importance for a large community and may help reconciling current discrepancies between measured RF of LAP in snow and chemical content measurements. However the paper suffers from several flaws that need, in my opinion, to be corrected before the paper can be published as described in my specific comments below.

**Specific comments**

1/ My first major comment is that the data and methods description lack a lot of details that are essential for the reader to understand correctly the results and conclusion of the paper :
- lines 84-101 : Please provide more details on how the sampling was performed. The snow samples are taken at the same time of the atmospheric sampling ? What is the volume ? To which snowpack depth does it correspond ?
- Lines 102-114 : Though the measurements methods are described in some other references, it would be very useful to add here the main concept, uncertainties and limitations
- Lines  115-124 : see comments 4
- Results and Discussion :
    - Whenever it's possible (description of Figures 3,5,7,9 and 10) please quantify the mean and std differences between the snow samples and the atmospheric samples
    - Figures 2, 4 and 8 : how was the classification performed ? Please explain in the methods part.
    - Line 161/162 : why are a-d representative of atmosphere ? And e-h of snow ?
    - Lines 195-197 : easily ??? please explain how (in the methods part), and add a reference.

2/ LAI is misleading (it also means Leaf Area Index). I would personally prefer the use of Ligth Absorbing Particles (LAPs) instead.

3/ The English is sometimes really difficult to understand and ambiguous. Though I am not an native English speaker, I would recommend a correction by a native English speaker.

4/ The RF change assessment is not detailed enough.

- Line 118-124, please describe again the conditions and parameters used in the simulations. It is required here even if it has been described previously in another paper. Why were such contents selected for the simulation ?
- Line 277 – 287, first describe the figure and the input for the different simulations. The difference in inputs is really difficult to guess

5/ Overall, I think the methods and data part should be largely extended to described in details all the methods used in the results part. In the results part, each Figure should be described first and then discussed or commented. The reading is really confusing otherwise.

**Minor comments**
Table 1 – Some spaces are missing (Sampling dates column)
Line 12 – "Aerosol impurities" this is quite redundant. Aerosols may be sufficient

Line 14 - "significantly varied " → will cause significant changes in radiative forcing

Everywhere : "glacier/snowpack" what do you exactly mean ? The snowpack on the glacier ? If yes, snowpack is probably sufficient.

Line 110-114. I don't understand this sentence. "Most" → please give a number.

Line 115 : "evaluated" → simulated

Line 160 : into → onto

Lines 186 : between the interfaces → between the snow and the atmosphere

Line 186-188 : very complicated and ambiguous sentence.

---

## Referee Comment (RC3) · J.-L. Ward (Referee) · 13 Oct 2018

Review of "Variability in individual particle structure and mixing states between the glacier snowpack and atmosphere interface in the northeast Tibetan Plateau" Dong et al., 2018.

The authors clearly show that the morphology of carbonaceous, dust, and other aerosol varieties changes between the atmosphere and the snowpack at all of their sampling locations in the Tibetan Plateau region. Their findings could significantly improve aerosol parameterizations in climate modeling environments, so I believe that this study is scientifically important and well-motivated.

I recommend this study for publication after major revisions. In my comments below, "L" means line. For example, L17 is "line 17".

MINOR:
    General:
        When you state acronyms for the first time, you must also define them. Please define the following:
            • SNICAR: L28, L81.
            • TSP: L87
            • DKL-2: L96
            • JEM-2100F: L106
    Introduction:
        After L82: the organization of the paper should be relayed to the readers here.

    Methods:
        L84-L86: Are these analysis varieties for atmospheric aerosols, terrestrial aerosols, or both? It's difficult to tell.

        L86-89: Is it possible for you to describe EDX, TSP, and TEM techniques in a couple of sentences? I'm a climate modeler, so I don't know anything about these methods.

        L89-L93: Because you have listed out all of the sampling locations in Table 1, I do not think you need to list those locations here. In lieu of this list, just refer the readers to Table 1.

        L95: What do you mean by "large-range"? Does this range refer to distance, or does it refer to time (taking measurements over long time spans)? The term is vague and should be changed to allow for easier interpretation.

        L100-L101: When you say the sampling method is similar to the Dong et al., 2017 study, do you use the same exact methods (are they identical?), or do you make small changes to these methods? If they are the exact same methods, you should say "the same as" instead of "similar to". If the methods are indeed "similar" but not identical, how are they different?

L113-114: What happened with samples not measured in frozen states? Did anything change about the methods? Or were all of the samples measured in frozen states? If so, make sure this is stated clearly.

L120: ug/g should be changed to μg/g.

Results:

L148: Use of "Meanwhile" is misleading. Please rephrase.

L152, L230: LAIs should be LAI.

L160-L162: You can delete the sentence starting with "Figure 4a-4d is representative of..." because this is mentioned in the caption for Figure 4.

L169-171: What previous work? Please cite this (these) reference (references) in this sentence.

L176-178: Since this sentence is basically the same information that is provided in Figure 5's caption, you can delete it.

L181-185: You stated that atmospheric BC/OM have higher ratios of fresh structure particles than the snowpack (L178-181). If this is what you are trying to say, you can delete the sentence starting with "We can demonstrate...". If you are trying to provide the readers with another result, please revise this sentence to make it clearer.

Radiative Forcing in other sections: either keep this information where it's at and delete Section 3.4, or wait to mention the following until Section 3.4:

> L188-192: You don't need to discuss the Peng et al. article here (or the fact that BC/OM particle structure changes lead to changes in radiative forcing on cyrospheric features). If you're keeping 3.4, move this to section 3.4.

> L216-219: (starts with "...many particles without salt-coating...") Move this to 3.4.

> L226, L236-L238: This discussion should be included in Section 3.4.

L196-197: What does "...on the advantage of the transmission micro-observation of the single particle structure" mean? This is unclear, so please revise the sentence to clearly depict what you are trying to say.

L197-199: Add "(Figure 6)" to the end of the sentence, and delete the sentence starting with "Figure 6 demonstrates...". The caption you have for Figure 6 provides the reader with this information.

L208-210: The sentence starting with "Figure 7 shows..." can be deleted since the caption for Figure 7 will provide the reader with this information. At the end of the previous sentence (L207-208), insert "(Figure 7)".

L264: "previous modeling studies"; which ones? Please cite the relevant sources in the document.

L265: "cell" (in phrase "cell core") is unclear. Do you mean "particle"?

L266-269: This is awkward and should be rewritten. State the studies you reference at the beginning of the sentence. Do all of the sources provide the same exact forcing values you cite? If not, provide the trends that the authors of all of the studies find between external and internal mixing. If the authors have markedly different results for internal versus external mixtures, their findings should be discussed in multiple sentences.

L269, L275: What do you mean by "heat-absorbing"? Would it be clearer to say "light-absorbing" instead?

L278-281: You're discussing the methods you use for SNICAR in a results section. This sentence should be part of the methods, not the results. Also, see below in the MAJOR revisions section for other questions I have regarding your SNICAR work.

Conclusions:

L300-302: If you keep Figure 11, either mention it in this sentence or delete "A schematic model diagram" and talk about how all of your results tie together.

L308-309: ("...the model...") you mean SNICAR, correct? If so, write SNICAR instead of "the model".

Figures and Tables:

Overall, I found the table and figures to be quite illustrative of your findings. Minor fixes/comments are listed below:

Figure 1: I recommend that the font color marking each sampling location be changed from black to white since black blends in with the topographical coloring scheme.

Figure 2: Does "mineral" mean dust in your microscopic images? If so, call it "dust" for clarity.

Figure 3: Is the photo (part c) really necessary? It seems like this would be something eye-catching for a presentation, but it doesn't really demonstrate any of your results.

Figure 4: The caption (L406) lists "Figures 3a-3d" and "Figures 3e-3h". These should be changed to "Figures 4a-4d" and "Figures 4e-4h", respectively.

Figure 5: Replace "Structure" with "LAI" (or something similar) in the caption (L408). "Structure" is a little ambiguous. Is part (a) the atmosphere and part (b) the snowpack? Revise your caption to show this.

Figure 8: (Caption) put a period at the end of the caption. Please label the panels.

Figure 9: (Caption) put a period at the end of the caption.

Figure 10: In the text, you refer to particle sizes on the order in MICROMETERS. In the figure, you use NANOMETERS (on the x-axis). Please be consistent; either change the figure to match to main text, or change the units you list in the main text.

>Also, the secondary y-axes on the right-hand side of the plot are color-coated to match mixing state, correct? Is it possible to redo either the axes or lines in the plot so that the colors more clearly match with the corresponding axes?

Figure 11: See my notes in the "Major Revisions" section below.

Figure 12: I think it would be more readable if the "Broadband Snow Albedo" values listed in the body of each plot are rounded to two decimal places. Also, please label each panel.

MAJOR:
   General:
      There are some grammatical errors that need to be addressed in this paper. The foremost issue I have found pertains to sentence structure. There are many instances of run-on sentences that can be split into multiple sentences and restructured. Examples of run-on sentences can be found in L15-L19, L22-25, L71-L79, and L120-L124. This list is not exhaustive, though, so you should check the entire manuscript (including figure captions) to find other grammatical errors.

   Methods:
      Which SNICAR configuration are you using? Are you using the online version? Or are you incorporating it into a climate model configuration? Please describe this in your methods section with 1-2 additional sentences.

   Results:
      Section 3.1 (shorter title would also be preferred): Is this supposed to be a summary of all of your findings? It is difficult to follow.

      Based on the title of the section, I think you should focus on the morphology of the particles. The reasons why the changes in particle morphology between the snow and atmosphere should be discussed in later sections. You could mention that these changes in morphology and structure lead to changes in radiative forcing, but that such impacts will be discussed in a later section.

L151: What does "aerosol change processes" mean? This is confusing phrasing that should be changed.

I interpret "aerosol change processes" as the changes in morphology and structure observed between aerosol species in the snow and in the atmosphere. If my interpretation is incorrect, what does "aerosol change processes" mean? Can you describe what information you are trying to relay here?

If my interpretation is correct, it seems as if you are saying that changes in morphology and structure (mostly through snow-based deposition) lead to large variability of individual LAI particle structures and morphology. This is redundant and need not be mentioned. If I am incorrect, then this sentence needs to be rewritten.

Section 3.2 (Again, the section title should be shorter) and Section 3.3
In the following instances, you are restating findings that have been previously discussed. Take one of the following steps: 1) If you are trying to say something new, rewrite the sentences, 2) If you are reiterating the point that is previously discussed, either a) delete the sentence (redundancy is not necessary) or justify why you want to restate this particular finding.

L186-188: What are you trying to say? Are you stating that dominantly fresh particles transitioned to aged particles from the atmosphere to the snow within the interface?

L212-214: Are you trying to say something new about salt coating of LAIs in the snowpack? If so, what information are you trying to provide to the reader?

Section 3.4: In the section heading, delete "Discussion of".

L241-243; L251-253: You've already stated these findings in previous sections. Since both of these findings are referenced to Figure 11, do you really need to include Figure 11 in the manuscript? In my mind, Figure 11 is not necessary for reader comprehension. Please justify why you wish to keep Figure 11 or delete it.

The discussion of your SNICAR-based findings (L277-287) should be at the beginning of this section. The first two paragraphs depict how your findings from previous sections match up with the literature, and what the literature suggests about how these findings will affect radiative forcing. Instead, use this information to justify why your SNICAR radiative forcing results make sense.

L283-L287: Although albedo reduction does imply positive radiative forcing, it would be convenient for readers to have access to calculated radiative forcing values (especially since this section is dedicated to radiative forcing).

I feel like this section is more of an afterthought (as it is currently written). However, the implications of morphology on snowmelt in the Tibetan Plateau region are important are directly related to your radiative forcing calculations. To better wrap up your findings, I think that the information you provide in this section should pertain more to your own calculations and less to the calculations of other authors.

Figures:

Figure 11: As I've asked above, is this really needed? You depict morphology, aging, and mixing changes in previous figures, and you state the radiative forcing tendencies in Section 3.4. The information depicted in this image represents the information that should be written up in the conclusion section (that is, it answers the following question: how are all of your findings connected?). Since you can easily describe how these conditions are connected, I do not think the figure is necessary.

---

## Author Comment (AC1) · 26 Nov 2018

Response to Anonymous Referee #1

General statement: This is an interesting study suitable for publication in TC. There are some important revisions required, but they should be easy for the authors to make.

Major comment: The large radiative forcing shown in Figure 11 (and cited in the abstract) is caused by the huge amounts of impurities in these glaciers. The authors need to point out that these glaciers are much more polluted than is normal for Tibetan glaciers. Line 120 gives the average value of BC (for 10 glaciers) as 854 ppm, or

854,000 ppb, i.e. about a factor of 40,000 larger than the amounts reported for Tibetan glaciers by Ming et al 2008 (ACP 8, 1343-1352). Some discussion is required before we can believe the results of this paper. I am also surprised that [BC] is~10 x [MD]. Previous reports find more MD than BC in Tibet.

Author response: Thank you. We have checked carefully here about the amount of BC and others, and we are sorry as it is a mistake by writing the wrong unit here. The unit should actually be ppb ($\mu$g /kg) for BC and OC, not ppm; for dust the unit is ppm ($\mu$g/g), as the average value of LAPs in the NTP region is derived from previous study (Zhang et al., 2017, 2018; Yan et al., 2016; and also Wang et al., 2013) for the snowpack. Thus the BC level is not that high.

We also checked carefully the result of the calculation using the model to calculate the albedo change based on the above data (see revised Figure 11). Moreover, we have checked throughout the paper, and we think the BC/dust level is still comparable to Ming et al. (2008, ACP) as their previous work result is derived from ice core, with relatively much higher average elevation in Everest (its deposition site with elevation 6500 m compared to 2900~4750 m a.s.l. of northeast Tibetan Plateau glacier sampling sites in this work) and lower atmospheric BC concentration. Besides, in this work we mainly focused on LAPs (BC, OC, mineral and others) in the glaciers and snowpacks for the surface distributed impurities, which is often accumulated in summer with surface melting and with higher BC concentration, and is thus actually different to that of ice core deposition record, but usually with higher mass level. (See Line 175-183 in the revision).

Besides, we need to clarify that the dust level is actually higher than that of BC in this work; here it is also caused by unit mistake, as the dust unit is $\mu$g /g, while the BC and OC unit here is $\mu$g/kg. The dust mass level we used here is actually much higher than BC (>10 times). Please also check that in the previous study in Table 2 of Zhang et al., 2018, TC.

Also see Figure 3 in this work, dust has a similar number concentration with BC in NTP region; however, dust is often larger particle and BC is often fine particles (often in PM2.5 and sub-micro section, Dong et al., 2016AE; Li et al., 2014 JGR), considering together the density of dust, thus BC is actually of smaller mass concentration than dust in the glacier/snowpack. Similar dust number concentration often means much higher mass concentration than BC; that is also why previous reports find more dust than BC in Tibet when compare mass concentration (Ming et al., 2008 ACP; Zhang et al, 2018 TC).

Moreover, in this study we mainly focus on individual LAPs particle mixing and structural change and its radiative forcing, thus we mainly use TEM-EDX method to calculate number concentration of the individual particle in the microscope observation. BC, OC, and dust mass data for the albedo simulation is derived from the average value of the previous study in the region (Zhang et al., 2017, 2018; Yan et al., 2016; and also Wang et al., 2013) for the snowpack result the northern Tibetan Plateau and also Qilian Mountains. See revised L170-171: In the model simulation, mineral dust ($93.2\pm27.05$ $\mu$g/g), BC ($1517\pm626$ $\mu$g/kg) and OC ($974\pm197\mu$g/kg) average concentration data, as well as other parameters. . .

References are also added to the revised manuscript: Ming J., Cachier H., Xiao C., et al., 2008. Black carbon record based on a shallow Himalayan ice core and its climatic implications, Atmos. Chem. Phys., 8, 1343–1352; Wang, X., Doherty, S., and Huang, J.: Black carbon and other light-absorbing impurities in snow across Northern China, J. Geophys. Res. Atmos., 118, 1471–1492, https://doi.org/10.1029/2012JD018291, 2013.

Minor Comments:

Line 43. "various salts . . . cause enhanced surface heat absorption". Which salts do you mean? Most salts are non-absorptive at UV, visible, and near-infrared wavelengths.

Author response: Yes, you are right, here we delete the salts, as salts in the atmosphere mainly influence the radiative forcing through salt-coating to BC/OC/dust, and also its hygroscopicity, which actually decreases the heat absorption (Li et al., 2014; Dong et al., 2017). See Line 46 in the revised manuscript.

Line 51. Delete "et al"

Author response: Yes, we deleted. Should be (Qiu, 2008)

Line 54. Anesio et al 2009 is missing from the reference list. Kaspari et al 2011 is also missing.

Author response: yes, we have added the references to the reference list in the revision. Anesio, A.M., Hodson, A.J., Fritz, A., et al., 2009. High microbial activity on glaciers: importance to the global carbon cycle. Global Change Biol. 15, 955-960. doi: 10.1111/j.1365-2486.2008.01758.x. Kaspari, S.D., Schwikowski, M., Gysel, M., et al., 2011. Recent increase in black carbon concentrations from a Mt. Everest ice core spanning 1860-2000 AD. Geophys. Res. Lett. 38, L04703 (2011).

Line 55. Xu et al. is missing from the reference list.

Author response: yes, we have added the reference to the reference list: Xu, B., et al. 2009. Black soot and the survival of Tibetan glaciers, Proc. Natl. Acad. Sci. U.S.A., 106(52), 22,114–22,118, doi:10.1073/pnas.0910444106.

Line 61. McConnell et al. 2007 is missing from the reference list.

Author response: yes, we revised, delete the reference here.

Line 74. Define TSP.

Author response: yes, revised, total suspended particle (TSP).

Line 105. Semeniuk et al. 2014 is missing from the reference list.

Author response: yes, revised: Semeniuk, T.A., Bruintjes, R.T., Salazar, V., Breed, D.W., Jensen, T.L., Buseck, P.R., 2014. Individual aerosol particles in ambient and

updraft conditions below convective cloud bases in the Oman mountain region. J. Geophys. Res. Atmos. 119, http://dx.doi.org/ 10.1002/2013JD021165.

Line 120. Please give the values of MD, BC, OC for each individual glacier. Put them in Table 1.

Author response: Thank you for the suggestion, we have provided the general average mass concentration of BC, OC, dust of snowpack and glaciers in the region as shown in the method section. Besides, the number concentration to the mineral dust, BC and OC based on TEM-EDX measurement has also been shown in Figure 3.

Line 169. "Previous work". Give a reference.

Author response: yes, revised, we add the reference here of (Peng et al., 2016; Yan et al., 2016). See revised Line 251.

Table 1. The altitude for DF is given as 390 m. Probably you mean 3900 m.

Author response: yes, it is a mistake and revised. See Table 1.

Table 1. Add three more columns, giving the concentrations of BC, OC, MD in the surface snow of each glacier.

Author response: we appreciate your suggestion; we have provided the general average mass concentration of BC, OC, dust of snowpack and glaciers in the region as shown in the method section. Besides, the number concentration to the mineral dust, BC and OC based on TEM-EDX measurement has also been shown in Figure 3.

Line 398 (Figure 2 caption) "nitrates". The legends in Figures 2 and 6 say nitrite not nitrate.

Author response: yes, we revised the Figures 2, 3 and 6. Here it should be nitrates in this work.

Line 400 (Figure 3 caption). "snow and ice". Which of the ten sites were snow; which

sites were ice?

Author response: yes, we revised. It should be the glacier and snowpack surface here, not exactly ice. We have described in the sampling section about glaciers and snowpack for the surface distributed impurities sampling. Here we revise to: Comparison of individual particles' compositions of light-absorbing impurities in the (a) atmosphere and (b) glacier and snowpack surface in the northeast Tibetan Plateau...(See revised Figure 3 caption in Line 518.)

Line 414 (Figure 8 caption) "mineral dust particles". No particles in Figure 8 are labeled as mineral dust.

Author response: yes, we revised. We delete the "mineral dust particles" in the caption.

Figure 2. Labels on the scale bars are illegible. Increase the font size.

Author response: yes, we revised to make it clear now. See revised Figure 2.

Figure 10. The vertical axes for the three graphs should all use the same scale, for easy comparison by the reader. Author response: yes, we revised. See the revised Figure 10.

Figure 10 legend. change "Mixted" to "Mixed".

Author response: yes, we revised.

Figure 12. The listed values for broadband albedo have too many significant figures. For example change "0.29774863" to "0.30".

Author response: yes, we revised. See revised Figure 11, as we change the order of Figure 11 and 12 based on the other review comments.

**Fig. 1.** revised Figure 2

[Figure]

**Fig. 2.** revised Figure 10

**Fig. 3.** revised Figure 11

---

## Author Comment (AC2) · 26 Nov 2018

Response to # Reviewer 2,

M. Dumont (Referee), marie.dumont@meteo.fr 2# Review of "Variability in individual particle structure and mixing states between glacier snowpack and atmosphere interface in the northeast Tibetan Plateau" by Dong et al.

Summary

This paper presents a dataset that explores the physical and chemical properties of light absorbing particles (LAPs) in the atmosphere and in the surface snowpack at sev-

eral places in the Tibetan Plateau. Observations from TEM and EDW measurements are described. A tentative scheme to explain the observations is proposed along with an assessment of the changes in radiative impact.

Recommendations

This is a really interesting, rich and fascinating dataset, the conclusions drawn by the authors are of importance for a large community and may help reconciling current discrepancies between measured RF of LAP in snow and chemical content measurements. However the paper suffers from several flaws that need, in my opinion, to be corrected before the paper can be published as described in my specific comments below.

Author response: Thank you for all the positive comments and suggestions. We have revised the manuscript very carefully based on your review comments.

Specific comments

1/ My first major comment is that the data and methods description lack a lot of details that are essential for the reader to understand correctly the results and conclusion of the paper:

lines 84-101: Please provide more details on how the sampling was performed. The snow samples are taken at the same time of the atmospheric sampling? What is the volume? To which snowpack depth does it correspond?

Author response: we revised by providing more details of sampling, and glacier/snowpack surface samples were collected on the glacier/snowpack surface (with 5 cm snow depth, each sampled for 200 mL) for comparison with the atmospheric deposition process, and the snow samples are taken at the same time of the atmospheric aerosol sampling, see revised Line 104-123:

During the fieldwork sampling, we used the middle-volume-sampler (DKL-2 with a flow rate of 150 L/min) for TEM filter sampling in this study, with a flow rate of 1 L min$-1$

were used for TSP filter sampling in our study, by a single-stage cascade impactor with a 0.5 mm diameter jet nozzle and an airflow rate of 1.0 L min-1. Each sample was collected with 1 hour duration. After collection, sample was placed in a sealed dry plastic tube and stored in a desiccator at 25°C and 20±3% RH to minimize exposure to ambient air before analysis, and particle smaller than 0.5 mm can be collected efficiently by the instruments. In total, 80 aerosol samples were collected directly on the calcium-coated carbon (Ca-C) grid filter. Additionally, 88 glacier/snowpack surface samples were collected on the glacier/snowpack surface (with 5 cm snow depth, each sampled for 200 mL) for comparison with the deposition process, and the snow samples are taken at the same time of the atmospheric aerosol sampling. The detailed aerosol/snow sampling method is similar to the previous study in Dong et al. (2016, 2017). The information on sampling location, time period and aerosol/snow sample number are shown in Table 1. Snow samples were collected at different elevations along the glacier surfaces of the study. Pre-cleaned low-density polyethylene (LDPE) bottles (Thermo scientific), stainless steel shovel, and super-clean clothes were used for the glacier/snowpack surface-snow sample collection. All samples were kept frozen until they were transported to the lab for analysis.

Lines 102-114 : Though the measurements methods are described in some other references, it would be very useful to add here the main concept, uncertainties and limitations.

Author response: Yes, we have revised. See the revised manuscript Line 130-146:

The analyses involved conventional and high-resolution imaging using bright field mode, electron diffraction (Semeniuk et al., 2014; Li et al., 2014), and energy-dispersive X-ray spectrometry. A qualitative survey of grids was undertaken to assess the size and compositional range of particles and to select areas for more detailed quantitative work that was representative of the entire sample. This selection ensured that despite the small percentage of particles analyzed quantitatively, our results were consistent with the qualitative survey of the larger particle population on each grid.

Quantitative information on size, shape, composition, speciation, mixing state, and physical state was collected for a limited set of stable particles. Volatile particles, including nitrate, nitrite, and ammonium sulfate, though not stable under the electron beam, can be detected on EDX at low beam intensity. EDX spectra were collected for 15 s in order to minimize radiation exposure and potential beam damage. All stable particles with sizes 20 nm to 35 um were analyzed within representative grid mesh squares located near the center of the grid. Grid squares with moderate particle loadings were selected for study to preclude the possibility of overlap or aggregation of particles on the grid after sampling. The use of Ca-C grids resulted in clear and unprecedented physical and chemical information for the individual particle types.

Lines 115-124 : See comments 4

Author Response: Yes, we revised the issue. See the response to comments 4

Results and Discussion :

Whenever it's possible (description of Figures 3, 5, 7, 9 and 10) please quantify the mean and std differences between the snow samples and the atmospheric samples

Author response: Yes, thank you for the suggestion; we revised through quantifying the mean and SD differences between the snow samples and the atmospheric samples in the revised manuscript as shown below.

which indicates the LAPs composition in atmosphere of various locations as BC (mean percentage of 18.3%, standard deviation (SD) 2.58), OC (28.2%, SD 3.49), NaCl (11%, SD 2.58), Sulfate (17%, SD 3.49); Ammonium (4.8%, SD 3.01), Nitrate (7%, SD 2.83), Mineral dust (13.7%, SD 3.02), whereas the LAPs composition in glacier/snowpack surface as: BC (mean 21.3%, SD 2.49), OC (31.2%, SD 2.44), NaCl (16.2%, SD 3.12), Sulfate (6.8%, SD 1.32), Ammonium (2%, SD 0.81), Nitrate (3.3%, SD 0.95), Mineral dust (19.2%, SD 2.9). (See revised manuscript Line 205-211).

Figure 5 indicates that in atmosphere the composition ratio is as fresh BC (mean percentage of 29.7%, with SD 3.95), fresh OC (41.8%, 4.34), aged BC (9.8%, 4.02), and aged OC (18.7%, 4.11); while in the snow the composition ratio is as fresh BC (mean percentage of 8.4%, SD 2.71), fresh OC (17.7%, 4.42), aged BC (31.5%, 2.99), and aged OC (42.4%, 4.45). (See revised manuscript Line 259-263).

In Figure 7, the salt-coated particles in atmosphere accounted for mean ratio 54.61% (with SD 12.02) in various locations, while that in snow of glacier/snowpack was 18.59% (SD 7.04). (See revised manuscript Line 291-293).

As shown in Figure 9, the internally mixed particles of BC in atmosphere accounted for mean ratio 4.68% (with SD 3.07) in various locations, whereas that in snow of glacier/snowpack was 14.85% (SD 4.93). (See revised manuscript Line 305-308).

In Figure 10, the mixings states of BC/OC in the glacier/snowpack snow of northeast Tibetan Plateau showed that the internally, single and externally mixed BC/OC individual particles account for a mean ratio of 69.2% (22.5), 5,35(1.72), and 25.95% (with SD 22.4), respectively. (See revised manuscript Line 315-318).

Figures 2, 4 and 8: How was the classification performed? Please explain in the methods part.

Author Response: We add a supplementary material for explaining the classification of each kind of individual aerosol particles. Please also see Table S1 in the revised manuscript.

Classification criteria of sampled particle types, mixing states and their possible sources in the snow/atmosphere samples were indicated in Table S1. (See Line 203-204 in the revised manuscript)

Table S1 Classification criteria of sampled particle types, mixing states and their possible sources in the snow/atmosphere samples.

Line 161/162: Why are a-d representative of the atmosphere? And e-h of snow?

Author response: we have revised to make the sentence clear:

Figure 4a-4d is the representative particles of fresh BC/OM with fractal morphology and a large amount in the atmosphere, whereas Figure 4e-4h is the representative particles of aged BC/OM with aggregated spherical morphology in the glacier/snowpack surface. (See revised manuscript Line 240-243)

Lines 195-197: easily? Please explain how (in the methods part), and add a reference.

Author response: We revised to explain the reason for salt-coating easily observation in TEM, as:

Using TEM-EDX microscope measurements, we can also easily derive the salt-coating conditions based on the advantage of the transmission observation to obtain individual particle inside-structure (Li et al., 2014). Particle (e.g. BC, OM) with salt coating will appear clearly surrounded by various salts shell and with the BC/OM particle as core (see revised manuscript Line 146-150).

Thus we also delete the similar sentence in section 3.3.

2/ LAI is misleading (it also means Leaf Area Index). I would personally prefer the use of Light Absorbing Particles (LAPs) instead.

Author response: yes, good suggestion, we revised LAI to Light Absorbing Particles (LAPs) throughout the revised manuscript.

3/ The English is sometimes really difficult to understand and ambiguous. Though I am not a native English speaker, I would recommend a correction by a native English speaker.

Author response: yes, we have revised carefully throughout the manuscript, and also improved the English language by Elsevier language editing service.

4/ The RF change assessment is not detailed enough.

Line 118-124, please describe again the conditions and parameters used in the simulations. It is required here even if it has been described previously in another paper. Why were such contents selected for the simulation?

Author response: yes, we have revised. See revised Line 158-192 in the revised manuscript.

We also simulated the albedo change contributed by individual particle mixing states' variability of LAPs. The SNICAR model can be used to simulate the albedo of snowpack by the combination of the impurity of the contents (e.g., BC, dust and volcanic ash), snow effective grain size, and incident solar flux parameters (Flanner et al., 2007). In the SNICAR model, the effective grain sizes of snow were derived from the stratigraphy and ranged from 100 $\mu$m for fresh clean snow to 1500 $\mu$m for aged snow and granular ice. The model was run with low, central, and high grain size for each snow type to account for the uncertainties in the observed snow grain sizes. Snow density varied with crystal size, shape, and the degree of rimming. The snow density data used in the SNICAR model are summarized with low-, central-, and high-density scenarios for the model run based on a series observations in the TP and previous literature (Judson and Doesken, 2000; Sjögren et al., 2007; Zhang et al., 2018). In the model simulation, mineral dust (93.2±27.05 $\mu$g/g), BC (1517±626 $\mu$g/kg) and OC (974±197 $\mu$g/kg) average concentration data, as well as other parameters, such as effective grain size, snow density, solar zenith angle, and snow depth on the glaciers, are considered, and mass absorption cross-sections (MAC) for salt-coated BC is referred to the average situation derived from the northern Tibetan Plateau glaciers (Zhang et al., 2017, 2018; Yan et al., 2016; Wang et al., 2013). Though showing high level, the BC concentration data used in this study is comparable to the previous work results derived from ice core (Ming et al., 2008), with relatively much higher average elevation in Everest (its deposition site elevation 6500 m compared to 2900-4750 m a.s.l. of northeast Tibetan Plateau glacier sampling sites) and lower atmospheric BC concentration. While in this work we mainly focus on LAPs (BC, OC, mineral and others) in the glaciers and snowpacks for the surface distributed impurities, which is often accumulated in summer with surface melting and with higher BC concentration.

When running the SNICAR model, BC was assumed to be coated or non-coated with sulfate (Flanner et al., 2007; Qu et al., 2014), or other salts. The mass absorption cross section (MAC) is an input parameter for the SNICAR model; it is commonly assumed to be 7.5 m2 /g at 550 nm for uncoated BC particles (Bond et al., 2013). For sulfate-coated BC particles, the MAC scaling factor was set to be 1 m2 /g, following Qu et al. (2014) and Wang et al. (2015). Other impurities (such as volcanic ash) were set to zero. In terms of the albedo calculation, RF due to BC and dust can be obtained by using Eq. (Kaspari et al., 2014; Yang et al., 2015):

Line 277-287, first describe the figure and the input for the different simulations. The difference in inputs is really difficult to guess

Author response: Yes, we have revised. See the revised last Results- section, revised manuscript Line 328-337.

Figure 12 showed the evaluation of snow albedo change of BC-salt coating change in the snowpack compared with that in the atmosphere using SNICAR model simulation in the MG, YG, QG, showing the albedo change of snow surface impurities in snowpack compared to that of the atmosphere. The parameters input for SNICAR model have been described in the method section. Mineral dust, BC and OC average concentration data, as well as other parameters, such as effective grain size, snow density, solar zenith angle, and snow depth on the glaciers, and MAC for BC were referred from the average situation in previous work of the northern Tibetan Plateau glaciers (Zhang et al., 2017, 2018; Yan et al., 2016; Wang et al., 2013).

Also see the revised Method section: Line 158-176, L185-196. We described the detailed input parameters for the model simulations for the LAPs in glacier snow.

We also simulated the albedo change contributed by individual particle mixing states'

variability of LAPs. The SNICAR model can be used to simulate the albedo of snow-pack by the combination of the impurity of the contents (e.g., BC, dust and volcanic ash), snow effective grain size, and incident solar flux parameters (Flanner et al., 2007). In this work, we use the online SNICAR model (http://snow.engin.umich.edu/). In the SNICAR model, the effective grain sizes of snow were derived from the stratigraphy and ranged from 100 $\mu$m for fresh clean snow to 1500 $\mu$m for aged snow and granular ice. The model was run with low, central, and high grain size for each snow type to account for the uncertainties in the observed snow grain sizes. Snow density varied with crystal size, shape, and the degree of rimming. The snow density data used in the SNICAR model are summarized with low-, central-, and high-density scenarios for the model run based on a series observations in the Tibetan Plateau and previous literature (Judson and Doesken, 2000; Sjögren et al., 2007; Zhang et al., 2018). In the model simulation, mineral dust (93.2$\pm$27.05 $\mu$g/g), BC (1517$\pm$626 $\mu$g/kg) and OC (974$\pm$197 $\mu$g/kg) average concentration data, as well as other parameters, such as effective grain size, snow density, solar zenith angle, and snow depth on the glaciers, were all considered; The mass absorption cross-sections (MAC) for salt-coated BC was referred to the average situation derived from the northern Tibetan Plateau glaciers (Zhang et al., 2017, 2018; Yan et al., 2016; Wang et al., 2013).

When running the SNICAR model, BC/OM was assumed to be coated or non-coated with sulfate (Flanner et al., 2007; Qu et al., 2014), or other salts. The mass absorption cross section (MAC) is an input parameter for the SNICAR model; it is commonly assumed to be 7.5 m2 /g at 550 nm for uncoated BC particles (Bond et al., 2013). For salt-coated BC particles, the MAC scaling factor was set to be 1 m2 /g, following Qu et al. (2014) and Wang et al. (2015). Other impurities (such as volcanic ash) were set to zero. In terms of the albedo calculation, the BC and dust radiative forcing (RF) can be obtained by using equation (1) (Kaspari et al., 2014; Yang et al., 2015):

5/ Overall, I think the methods and data part should be largely extended to describe in details all the methods used in the results part. In the results part, each Figure should

be described first and then discussed or commented. The reading is really confusing otherwise.

Author response: Yes, we have revised according to your suggestion. Also, see the detailed response to the above-related issues.

Minor comments

Table 1 – Some spaces are missing (Sampling dates column)

Author response: Yes, we revised. See revised Table 1.

Line 12 – "Aerosol impurities" this is quite redundant. Aerosols may be sufficient

Author response: Yes, we revised. "Aerosol impurities" to "aerosols".

Line 14 - "significantly varied" → will cause significant changes in radiative forcing

Author response: Yes, we revised.

Everywhere: "glacier/snowpack" what do you exactly mean? The snowpack on the glacier? If yes, snowpack is probably sufficient.

Author response: Yes, we revised. We revise to "glacier and snowpack surfaces" here, as here we mean both the glacier surface snow and snow cover/snowpack samples in the high areas.

Line 110-114. I don't understand this sentence. "Most" → please give a number.

Author response: Yes, we revised. It is writing mistake here.

Moreover, as the snow samples melting will affect the individual particle composition during the measurements, especially for various types of salts because the salt is unstable in high temperature (e.g. Ammonium and Nitrates) and will change, thus the snow/aerosol samples were directly observed under the TEM instrument and measured before it melted. All samples were measured in frozen states. (Line 152-157 in the revised manuscript)

Line 115: "evaluated" → simulated

Author response: Yes, we revised.

Line 160: into → onto

Author response: Yes, we revised.

Lines 186: between the interfaces → between the snow and the atmosphere

Author response: Yes, we revised.

Line 186-188: very complicated and ambiguous sentence.

Author response: Yes, we revised, as suggested by another reviewer the sentence is redundant with a similar meaning of the previous sentence. Thus we delete the sentence here.

Please also note the supplement to this comment:
https://www.the-cryosphere-discuss.net/tc-2018-166/tc-2018-166-AC2-supplement.pdf

**Supplement:**

**Table S1 Classification criteria of sampled particle types, mixing states and their possible sources in the snow/atmosphere samples**

| Particle types | Featured element composition | Mixing properties | Sources | References |
|---|---|---|---|---|
| Mineral dust | Si, Al, Fe, Ca, Mg-rich, such as clay, quartz, feldspar, albite, with minor calcite, and other oxides. | Reacted minerals aggregated with soot and salt (MCS, nitrite, etc.). | Desert sand and crustal surface soil. | Shao et al., 2007 Laskin et al., 2005 Dong et al., 2017; |
| Soot (BC) | C (dominant) and O-rich. | C-rich materials mixed with organic, S, and K-rich particles. | Fossil fuels and biomass burning. | Li et al., 2014 |
| Fly ash | Si, Al, Fe, S, and Ti-rich. | Fly ash mixed with salt (NaCl, sulfate), metals ($Fe_2O_3$, $MnO_2$), silicate containing minor Fe, Mn, Ti and other metals. | Coal-fired power plants, heavy industries, and oil refinery. | Shi et al., 2003 Li et al., 2014 |
| Organic matter | C (dominant), O, Si-rich, and regular spherical organic particles. | Mixed with mineral, S-rich and K-rich pollutants particles. | Biological particle, fossil fuels and biomass burning. | Hand et al., 2005; Chakrabarty et al., 2006 |
| Sulfate | S- (dominant) rich and mixing sulfate cation (K, Ca, Na, and Mg). | Mixed cation sulfate, $(HN_4)_2SO_4$, and often coated with mineral, soot, and organic particles. | Fossil fuels emission and secondary particles formed by $SO_2$ and $NO_x$. | Li et al., 2014 Li and Shao, 2009b Niemi et al., 2006; |
| Nitrite | N (dominant), O, K, and Na-rich. | Coated and mixed with other type particles (sulfate, mineral, soot, and organic). | Fossil fuels and secondary particles formed by $NO_x$. | Niemi et al., 2006 Adachi and Buseck, 2008 |
| NaCl | NaCl rich salt. | Cubic NaCl particle, often coated by $NaNO_3$ and $Na_2SO_4$. | Sea salt from the Indian Ocean and other seas, salt from arid dust regions. | Li et al., 2014 Vester et al.,2007 |
| Ammonium | $(HN_4)_2SO_4$ and $(HN_4)_2NO3$. | Mixed with MCS, nitrite and minerals. | Fossil fuels and secondary particles formed $NH_3$. | Li et al., 2014 |

---

## Author Comment (AC3) · 26 Nov 2018

Response to Reviewer3,

jamiewa@umich.edu,

Review of "Variability in individual particle structure and mixing states between the glacier snowpack and atmosphere interface in the northeast Tibetan Plateau" Dong et al., 2018.

The authors clearly show that the morphology of carbonaceous, dust, and other aerosol varieties changes between the atmosphere and the snowpack at all of their sampling lo-

cations in the Tibetan Plateau region. Their findings could significantly improve aerosol parameterizations in climate modeling environments, so I believe that this study is scientifically important and well-motivated. I recommend this study for publication after major revisions. In my comments below, "L" means line. For example, L17 is "line 17".

MINOR:

General:

When you state acronyms for the first time, you must also define them. Please define the following:

SNICAR: L28, L81.

Author response: Yes, we revised: Single-layer implementation of the Snow, Ice, and Aerosol Radiation (SNICAR) model. See revised Line 84 in the revision.

TSP: L87

Author response: total suspended particle (TSP), see revised Line 77 in the revised manuscript.

DKL-2: L96

Author response: Here DKL is Dankeli in Chinese, which means individual particles. See revised Line 105 in the revision.

JEM-2100F: L106

Author response: Japan Electron Microscope (JEM), See revised Line 129 in the revision.

Introduction:

After L82: the organization of the paper should be relayed to the readers here.

Author response: Yes, we revised by adding the organization of the paper:

We organized the paper as follows: In section 2, we provided detailed descriptions about data and method of individual aerosol particle sampling and analysis; and in section 3 we presented the observed results and discussion of: (i) comparison of LAPs components between atmosphere and snowpack interface; (ii) BC/OM particle aging variability between atmosphere and snowpack interface; (iii) changes in salt-coating conditions and BC/OM mixing states; (iv) particle mixing states variability and its contribution to light absorbing. In section 4, we concluded our results and also provided the future study objective for the community. (See the revised manuscript Line 86-94.)

Methods:

L84-L86: Are these analysis varieties for atmospheric aerosols, terrestrial aerosols, or both? It's difficult to tell.

Author response: Yes, this is about both atmospheric aerosols and also snowpack LAPs, as we indicated in this section:

Atmospheric LAPs samples (TEM aerosol filter samples) and the glacier/snowpack surface distributed impurity samples were both collected across the northeastern Tibetan Plateau region in summer between June 2016 and September 2017. (Line 98-100 in the revised manuscript)

Ca-C grids were used as filters with the advantage of clear and unprecedented observation for single-particle analyses of aerosols and snowpack samples (Line 125-127)

L86-89: Is it possible for you to describe EDX, TSP, and TEM techniques in a couple of sentences? I'm a climate modeler, so I don't know anything about these methods.

Author response: Yes, thank you for the suggestion. We revised these methods with more description.

TSP:

During the fieldwork sampling, we used the middle-volume-sampler (DKL-2 with a flow

rate of 150 L/min) for TEM filter sampling in this study, with a flow rate of 1 L min−1 were used for TSP filter sampling in our study, by a single-stage cascade impactor with a 0.5 mm diameter jet nozzle and an airflow rate of 1.0 L min-1. Each sample was collected with 1 hour duration. After collection, sample was placed in a sealed dry plastic tube and stored in a desiccator at 25°C and 20±3

TEM-EDX:

A qualitative survey of grids was undertaken to assess the size and compositional range of particles and to select areas for more detailed quantitative work that were representative of the entire sample. This selection ensured that despite the small percentage of particles analyzed quantitatively, our results were consistent with the qualitative survey of the larger particle population on each grid. Quantitative information on size, shape, composition, speciation, mixing state, and physical state was collected for a limited set of stable particles. Some LAPs particles, including nitrate, nitrite, and ammonium sulfate, though not stable under the electron beam, can be well detected on EDX at low beam intensity. EDX spectra were collected for 15 s in order to minimize radiation exposure and potential beam damage. All stable particles with sizes 20 nm to 35 $\mu$m were analyzed within representative grid mesh squares located near the center of the grid. Grid squares with moderate particle loadings were selected for study to preclude the possibility of overlap or aggregation of particles on the grid after sampling. The use of Ca-C grids resulted in clear and unprecedented physical and chemical information for the individual particle types. Using TEM-EDX microscope measurements, we can also easily derive the salt-coating conditions based on the advantage of the transmission observation to obtain individual particle inside-structure. Particle (e.g. BC, OM) with salt coating will appear clearly surrounded by various salts shell and with the BC/OM particle as core. (Line 132-150 in the revised manuscript)

L89-L93: Because you have listed out all of the sampling locations in Table 1, I do not think you need to list those locations here. In lieu of this list, just refer the readers to Table 1.

Author response: Yes, we delete the details of the locations, and revised as:

Figure 1 shows the sampling locations and their spatial distribution in the region, including locations in the eastern Tianshan Mountains, the Qilian Mountains, the Kunlun Mountain and the Hengduan Mountains, where large-range spatial scale observations were conducted (see Table 1). (Line 101-104 in the revised manuscript)

L95: What do you mean by "large-range"? Does this range refer to distance, or does it refer to time (taking measurements over long time spans)? The term is vague and should be changed to allow for easier interpretation.

Author response: Yes, we revised as:

. . .where large-range spatial scale observations were conducted. (Line 104)

L100-L101: When you say the sampling method is similar to the Dong et al., 2017 study, do you use the same exact methods (are they identical?), or do you make small changes to these methods? If they are the exact same methods, you should say "the same as" instead of "similar to". If the methods are indeed "similar" but not identical, how are they different?

Author response: Yes, we used the same method with our previous study and we revise to: The aerosol/snow sampling method is also same to the previous study in Dong et al. (2016, 2017). The detailed information on sampling location, time period and aerosol/snow sample number are shown in Table 1. Snow samples were collected at different elevations along the glacier/snowpack surfaces of the study. Pre-cleaned low-density polyethylene (LDPE) bottles (Thermo scientific), stainless steel shovel, and super-clean clothes were used for the glacier/snowpack surface-snow sample collection. All samples were kept frozen until they were transported to the lab for analysis. (See revised Line 116-123 in the revised manuscript)

L113-114: What happened with samples not measured in frozen states? Did anything change about the methods? Or were all of the samples measured in frozen states? If

so, make sure this is stated clearly.

Author response: Yes, we revised this as below:

Moreover, as the snow samples melting will affect the individual particle composition during the measurements, especially for various types of salts because the salt is unstable in high temperature (e.g. Ammonium and Nitrates) and will change, thus the snow/aerosol samples were directly observed under the TEM instrument and measured before it melted. All samples were measured in frozen states. (See revised Line 152-157 in the revised manuscript).

L120: ug/g should be changed to $\mu$g/g.

Author response: Yes, we revised. See revised Line 170-171.

Results:

L148: Use of "Meanwhile" is misleading. Please rephrase.

Author response: Yes, we revised to: Moreover

L152, L230: LAIs should be LAI.

Author response: thank you, we have revised the LAIs to LAPs throughout the paper, as light-absorbing particles (LAPs)

L160-L162: You can delete the sentence starting with "Figure 4a-4d is representative of..." because this is mentioned in the caption for Figure 4.

Author response: Yes, thank you, we keep the sentence for clarify here as another review suggested.

L169-171: What previous work? Please cite this (these) reference (references) in this sentence.

Author response: we added the references:

[Figure]

Previous work has indicated the structure and mass absorption cross (MAC) section change of BC particles in the atmosphere (Peng et al., 2016; Yan et al., 2016). See revised manuscript Line 252.

L176-178: Since this sentence is basically the same information that is provided in Figure 5's caption, you can delete it.

Author response: Yes, thank you, we keep the sentence for clarify here. As another reviewer suggest we describe the figure first in the text and then discuss. And it also will help to clarify. Thus we keep the sentence.

L181-185: You stated that atmospheric BC/OM have higher ratios of fresh structure particles than the snowpack (L178-181). If this is what you are trying to say, you can delete the sentence starting with "We can demonstrate..."; If you are trying to provide the readers with another result, please revise this sentence to make it clearer.

Author response: Yes, we deleted the sentence, see Line 267-273.

The amount of aged particles in snowpack is 2-3 times higher than that in the atmosphere. In the atmosphere the BC/OM both showed high ratios of fresh structure particles (fractal morphology), while in the glacier/snowpack surface more particles indicated aged structure (spherical morphology), although there were a small portion of particles still fresh (Figure 5). The change proportion of BC/OM particle aging is very marked between the snow and the atmosphere.

Radiative Forcing in other sections: either keep this information where it's at and delete Section 3.4, or wait to mention the following until Section 3.4:

Author response: Yes, we revise the radiative forcing to light absorbing, and heat absorbing here (see Line 273 and Line 275 in the revision). And we mainly discussed the radiative forcing influence in section 3.4.

L188-192: You don't need to discuss the Peng et al. article here (or the fact that BC/OM particle structure changes lead to changes in radiative forcing on cryospheric features).

If you're keeping 3.4, move this to section 3.4.

Author response: Thank you for suggestion, here we delete "as shown in previous studies", and keep the main words to show the environmental and climatic importance of the particle structure change. See revised Line 273-277 in the revised manuscript.

L216-219: (starts with "...many particles without salt-coating...") Move this to 3.4.

Author response: Yes, we move the sentence to 350-353 in section 3.4, see the revised manuscript. L226, L236-L238: This discussion should be included in Section 3.4.

Author response: Yes, we delete and move this to the 3.4 section. See Line 375-379 in the revised manuscript.

L196-197: What does "...on the advantage of the transmission micro-observation of the single particle structure" mean? This is unclear, so please revise the sentence to clearly depict what you are trying to say.

Author response: Yes, We delete the sentence here and move to the method section:

Using TEM-EDX microscope measurements, we can also easily derive the salt-coating conditions based on the advantage of the transmission observation to obtain individual particle inside-structure. Particle (e.g. BC, OM) with salt coating will appear clearly surrounded by various salts shell and with the BC/OM particle as core. (See Line 146-150 in the revised manuscript).

L197-199: Add "(Figure 6)" to the end of the sentence, and delete the sentence starting with "Figure 6 demonstrates...". The caption you have for Figure 6 provides the reader with this information.

Author response: Yes, thank you. We add the Figure 6 to the end of the sentence. As another reviewer suggested that in the results part, each Figure should be described first and then discussed or commented, thus we also keep the sentence.

L208-210: The sentence starting with "Figure 7 shows..." can be deleted since the

caption for Figure 7 will provide the reader with this information. At the end of the previous sentence (L207-208), insert "(Figure 7)".

Author response: Yes, we add Figure 7 to the end of the sentence. Similarly, we also keep the sentence.

L264: "previous modeling studies"; which ones? Please cite the relevant sources in the document.

Author response: Yes, we add the references:

Internally mixed particles of BC/OM have showed the strongest light absorption in previous modeling studies (Cappa et al., 2012; Jacobson et al., 2000). (see Line 370-371 in the revised manuscript)

L265: "cell" (in phrase "cell core") is unclear. Do you mean "particle"?

Author response: Yes, cell here means "cell particle" with organic matter coating. We revise to: BC acts as a cell-core particle. (Line 371 in the revised manuscript)

L266-269: This is awkward and should be rewritten. State the studies you reference at the beginning of the sentence. Do all of the sources provide the same exact forcing values you cite? If not, provide the trends that the authors of all of the studies find between external and internal mixing. If the authors have markedly different results for internal versus external mixtures, their findings should be discussed in multiple sentences.

Author response: Yes, we revised as below:

In previous study the mixing state was found to affect the BC global direct forcing by a factor of 2.9 (0.27 Wm-2 for an external mixture, +0.54 Wm-2 for BC as a coated core, and +0.78 Wm-2 for BC as well mixed internally) (Jacobson, 2000), and that the mixing state and direct forcing of the black-carbon component approach those of an internal mixture, largely due to coagulation and growth of aerosol particles (Jacobson et al.,

2001), and also found radiative absorption enhancements due to the mixing state of BC as indicated in Cappa et al. (2012) and He et al.(2015). (See revised Line 372-379).

L269, L275: What do you mean by "heat-absorbing"? Would it be clearer to say "light-absorbing" instead?

Author response: Yes, we revised to light-absorbing in the revised manuscript.

L278-281: You're discussing the methods you use for SNICAR in a results section. This sentence should be part of the methods, not the results. Also, see below in the MAJOR revisions section for other questions I have regarding your SNICAR work.

Author response: Yes, that is right. Here we just provide the description and to clarify its relation with the LAPs particle in this work. Please also see the related response in the below issue.

Conclusions:

L300-302: If you keep Figure 11, either mention it in this sentence or delete "A schematic model diagram" and talk about how all of your results tie together.

Author response: Yes, we keep the Figure 12 (the original Figure 11, as we revise the figure order) in the revised manuscript to describe the change and give a total model of this work, thus we revised to: A schematic model diagram shown in the figure linking the explanation the LAPs' structure aging and salt-coating change and comparing their influences to the radiative forcing between the atmosphere and glacier snowpack was presented in the study. The LAPs in glacier/snowpack will change to more aged and internally mixed states compared to that of atmosphere. Thus, the light absorption of the LAPs as a whole will increase greatly in glacier snowpack environments. (Line 400-405 in the revised manuscript)

L308-309: ("...the model...") you mean SNICAR, correct? If so, write SNICAR instead of "the model".

[Figure]

Author response: Yes, we revised to the SNICAR modeling. See the revised manuscript.

Figures and Tables:

Overall, I found the table and figures to be quite illustrative of your findings. Minor fixes/comments are listed below:

Figure 1: I recommend that the font color marking each sampling location be changed from black to white since black blends in with the topographical coloring scheme.

Author response: Yes, we revised. See revised Figure 1.

Figure 1 Location map showing the sampled glaciers and snowpack in the northeast Tibetan Plateau, including the Miaoergou Glacier (MG), Laohugou Glacier No.12 (LG12), Qiyi Glacier (QG), Lenglongling Glacier (LG), Shiyi Glacier (SG), Dabanshan snowpack (DS), Yuzhufeng Glacier (YG), Gannan Snowpack (GS), Dagu Glacier (DG), and Hailuogou Glacier (HG), where large-range field observations of atmosphere and glacier surface impurities were conducted.

Figure 2: Does "mineral" mean dust in your microscopic images? If so, call it "dust" for clarity. Author response: Yes, thank you. Mineral means mineral dust here, as mineral is more correct here for the LAPs components analysis in the TEM-EDX measurement, and dust may contain mineral dust and anthropologic source dust, thus we keep as mineral in this figure. And we revise in the caption to mineral dust.

Figure 3: Is the photo (part c) really necessary? It seems like this would be something eye-catching for a presentation, but it doesn't really demonstrate any of your results.

Author response: Yes, we here use the photo to present the large range snow-cover environment in the northeast Tibetan Plateau region; thereby it is important to the radiative forcing and climate with the LAPs deposition in the snowpack from atmosphere.

Figure 4: The caption (L406) lists "Figures 3a-3d" and "Figures 3e-3h". These should

[Figure]

be changed to "Figures 4a-4d" and "Figures 4e-4h", respectively.

Author response: Yes, it is a mistake, we revised. See revised Figure 4 caption.

Figure 5: Replace "Structure" with "LAI" (or something similar) in the caption (L408). "Structure" is a little ambiguous. Is part (a) the atmosphere and part (b) the snowpack? Revise your caption to show this.

Author response: Yes, we revised to LAPs here. See the revised Figure 5 caption:

Figure 5 LAPs aging of BC/OC individual impurity particles and composition ratio (

Figure 8: (Caption) put a period at the end of the caption. Please label the panels.

Author response: Yes, we revised; see revised Figure 8 and caption.

Figure 8 Internal mixing states of BC (soot) and OM in the various glacier snowpack in northeast Tibetan Plateau in summer 2016-2017

Figure 9: (Caption) put a period at the end of the caption.

Author response: Yes, we revised. Figure 9 The proportion change of internally mixed BC particle with other particlesïijŇshowing the obvious increase of internally mixed BC/OM in glacier snowpack compared with those in the atmosphere in summer 2016-2017

Figure 10: In the text, you refer to particle sizes on the order in MICROMETERS. In the figure, you use NANOMETERS (on the x-axis). Please be consistent; either change the figure to match to main text, or change the units you list in the main text. Also, the secondary y-axes on the right-hand side of the plot are color-coated to match mixing state, correct? Is it possible to redo either the axes or lines in the plot so that the colors more clearly match with the corresponding axes?

Author response: Yes, we revised to micrometer ($\mu$m). We have checked the y-axis, and make it to one y-axis in the revised figure. Please see revised Figure 10.

Figure 10 Average conditions of single, internally and externally mixed BC/OM individual particles in the snowpack of northeast Tibetan Plateau glaciers, showing most of the BC/OM with diameter >1 $\mu$m in internally mixing conditions.

Figure 11: See my notes in the "Major Revisions" section below.

Author response: yes, revised. See the response to the section.

Figure 12: I think it would be more readable if the "Broadband Snow Albedo" values listed in the body of each plot are rounded to two decimal places. Also, please label each panel.

Author response: Yes, we revised. As we change the order of Figure 11 and Figure 12 in the revised manuscript, thus see revised Figure 11.

Figure 11 Evaluation of snow albedo change of BC-salt coating change in the snowpack compared with atmosphere using SNICAR model simulation in the MG (a, b), YG (c, d), LG 12 (e, f), which shows the largely decreased albedo of snow surface impurities in snowpack compared to that of the atmosphere, implying markedly enhanced radiative forcing in the snowpack surface impurities.

MAJOR:

General:

There are some grammatical errors that need to be addressed in this paper. The foremost issue I have found pertains to sentence structure. There are many instances of run-on sentences that can be split into multiple sentences and restructured. Examples of run-on sentences can be found in L15-L19, L22-25, L71-L79, and L120-L124. This list is not exhaustive, though, so you should check the entire manuscript (including figure captions) to find other grammatical errors.

Author response: Yes, we have checked and revised very carefully all the grammatical errors throughout the manuscript.
Methods:

Which SNICAR configuration are you using? Are you using the online version? Or are you incorporating it into a climate model configuration? Please describe this in your methods section with 1-2 additional sentences.

Author response: yes, thank you. We here used the online SNICAR model, we revised: In this work we use the online SNICSR model (http://snow.engin.umich.edu/). (Line 162-163 in the revised manuscript)

Results:

Section 3.1 (shorter title would also be preferred): Is this supposed to be a summary of all of your findings? It is difficult to follow.

Based on the title of the section, I think you should focus on the morphology of the particles. The reasons why the changes in particle morphology between the snow and atmosphere should be discussed in later sections. You could mention that these changes in morphology and structure lead to changes in radiative forcing, but that such impacts will be discussed in a later section.

Author response: Yes, thank you, this section mainly focuses on the morphology and components of the particles, and also its change between the atmosphere and snow. Thus we revise to make the title shorter as "3.1 Comparison of LAPs Components between Atmosphere and Snowpack Interface".

As we have revised the subtitles and arrange the contents in section 3 as following: (i) comparison of LAPs components between atmosphere and snowpack interface; (ii) BC/OM particle aging variability between atmosphere and snowpack interface; (iii) changes in salt-coating conditions and BC/OM mixing states; (iv) particle mixing states variability and its contribution to light absorbing. Thus this section 3.1 will be the main description of LAPs components and its change between the atmosphere and snow interface, and also its related reason for the variability, which is a general components

situation of LAPs in snow and atmosphere. The next following 3 parts (3.2, 3.3, and 3.4) are mainly about the particle aging, salt coating and its radiative influence, respectively.

We also revised to clarify that the influence of such change will be discussed in later section in the revised manuscript: The change in morphology and structure will undoubtedly cause a significant variability of impurities' heat absorbing property in both the atmosphere and the glacier/snowpack surface, and such impacts will be discussed in a later section. (Line 226-228 in the revised manuscript)

L151: What does "aerosol change processes" mean? This is confusing phrasing that should be changed. I interpret "aerosol change processes" as the changes in morphology and structure observed between aerosol species in the snow and in the atmosphere. If my interpretation is incorrect, what does "aerosol change processes" mean? Can you describe what information you are trying to relay here? If my interpretation is correct, it seems as if you are saying that changes in morphology and structure (mostly through snow-based deposition) lead to large variability of individual LAI particle structures and morphology. This is redundant and need not be mentioned. If I am incorrect, then this sentence needs to be rewritten.

Author response: yes, thank you, we revised the sentence as below:

Aerosol LAPs change during the atmospheric transport and deposition processes (especially through wet deposition with precipitating-snow) will mainly lead to large variability of individual particle's structure and morphology. (See revised Line 231-233)

Section 3.2 (Again, the section title should be shorter) and Section 3.3

In the following instances, you are restating findings that have been previously discussed. Take one of the following steps: 1) If you are trying to say something new, rewrite the sentences, 2) If you are reiterating the point that is previously discussed, either a) delete the sentence (redundancy is not necessary) or justify why you want to restate this particular finding.

L186-188: What are you trying to say? Are you stating that dominantly fresh particles transitioned to aged particles from the atmosphere to the snow within the interface?

Author response: yes, it is really somehow repetition and redundant, with similar meaning as the previous sentence. Thus we delete the sentence here in the revised manuscript.

L212-214: Are you trying to say something new about salt coating of LAIs in the snow-pack? If so, what information are you trying to provide to the reader?

Author response: yes, thank you, it is also redundant, with similar meaning as the previous sentence. Thus we delete the sentence here in the revised manuscript.

Section 3.4: In the section heading, delete "Discussion of".

Author response: yes, we delete and change the title to: "3.4 Particle Mixing States Variability and Its Contribution to Light Absorbing. "

L241-243; L251-253: You've already stated these findings in previous sections. Since both of these findings are referenced to Figure 11, do you really need to include Figure 11 in the manuscript? In my mind, Figure 11 is not necessary for reader comprehension. Please justify why you wish to keep Figure 11 or delete it.

Author response: Yes, we here presented the Figure 11 to show the schematic diagram model for the explanation of the particle structure aging and salt-coating changes, and a comparison of its influence to the radiative forcing, which is a synthesis of the study to reveal the individual LAPs change between the two kinds of medium (snow and atmosphere), and its final influence to the radiative and climate change. Thus we think it is very helpful to interpret visually the thesis of this work. We also revised the Figure 11 to make this point clearer. Please also see the revised Figure 12 (please notice that the figure order changes in the revision).

Figure 12 Schematic diagram linking aging and salt coating change and comparing its influence to the radiative forcing between the atmosphere and snowpack of a remote

glacier basin, causing markedly enhanced heat absorption.

The discussion of your SNICAR-based findings (L277-287) should be at the beginning of this section. The first two paragraphs depict how your findings from previous sections match up with the literature, and what the literature suggests about how these findings will affect radiative forcing. Instead, use this information to justify why your SNICAR radiative forcing results make sense.

Author response: Yes, that is good suggestion; we have revised the order of Figure 11 and Figure 12. And the discussion order of SNICAR modeling and the general Schematic diagram were also changed in the revised manuscript. See revised Section 3.4.

L283-L287: Although albedo reduction does imply positive radiative forcing, it would be convenient for readers to have access to calculated radiative forcing values (especially since this section is dedicated to radiative forcing). I feel like this section is more of an afterthought (as it is currently written). However, the implications of morphology on snowmelt in the Tibetan Plateau region are important are directly related to your radiative forcing calculations. To better wrap up your findings, I think that the information you provide in this section should pertain more to your own calculations and less to the calculations of other authors.

Author response: Yes, good suggestion. We revised this section as you suggested and evaluated the radiative forcing based on the albedo calculation, and revised the manuscript as below:

Based on the LAPs salt-coating-induced albedo changes, RF was calculated by equation (1) for the different scenarios. The results show that the RF caused by salt coating changes, varied between 1.6–26.3 W m2 depending on the different scenarios (low, central, and high snow density), respectively. (See Line 339-343 in the revised manuscript)

Figures:

Figure 11: As I've asked above, is this really needed? You depict morphology, aging, and mixing changes in previous figures, and you state the radiative forcing tendencies in Section 3.4. The information depicted in this image represents the information that should be written up in the conclusion section (that is, it answers the following question: how are all of your findings connected?). Since you can easily describe how these conditions are connected, I do not think the figure is necessary.

Author response: yes, we think the Figure 11 is necessary in this work; we revised the figure to improve this meaning more complete for readers and a general LAPs change model between the atmosphere and snowpack (see revised Figure 12, the figure order changes). We appreciate your comments, as discussed above; we think Figure 11 is very helpful to interpret visually the thesis of this work. Besides, this part discussion is also an improvement of the whole paper, by putting the entire factors together to see its total influence.
* * *
[Figure]

[Figure]

Fig. 1. revised Figure 1

**a** BC OM 100 nm

**b** BC OM 200 nm

**c** BC OM 500 nm

**d** BC OM 200 nm

**e** BC OM 500 nm

**f** OM BC OM 100 nm

**Fig. 2.** revised Figure 8

[Figure]

**Fig. 3.** revised Figure 10

Fig. 4. revised Figure 11

**Fig. 5.** revised Figure 12